# Ezrin enhances line tension along transcellular tunnel edges via NMIIa driven actomyosin cable formation

Caroline Stefani[1,2], David Gonzalez-Rodriguez[3], Yosuke Senju[4], Anne Doye[1], Nadia Efimova[5], Sébastien Janel[6], Justine Lipuma[1], Meng Chen Tsai[1], Daniel Hamaoui[1], Madhavi P. Maddugoda[1], Olivier Cochet-Escartin[7,8], Coline Prévost[7,8], Frank Lafont[6], Tatyana Svitkina[5], Pekka Lappalainen[4], Patricia Bassereau[7,8] & Emmanuel Lemichez[1,9]

Transendothelial cell macroaperture (TEM) tunnels control endothelium barrier function and are triggered by several toxins from pathogenic bacteria that provoke vascular leakage. Cellular dewetting theory predicted that a line tension of uncharacterized origin works at TEM boundaries to limit their widening. Here, by conducting high-resolution microscopy approaches we unveil the presence of an actomyosin cable encircling TEMs. We develop a theoretical cellular dewetting framework to interpret TEM physical parameters that are quantitatively determined by laser ablation experiments. This establishes the critical role of ezrin and non-muscle myosin II (NMII) in the progressive implementation of line tension. Mechanistically, fluorescence-recovery-after-photobleaching experiments point for the upstream role of ezrin in stabilizing actin filaments at the edges of TEMs, thereby favouring their crosslinking by NMIIa. Collectively, our findings ascribe to ezrin and NMIIa a critical function of enhancing line tension at the cell boundary surrounding the TEMs by promoting the formation of an actomyosin ring.

[1] INSERM, U1065, Université de Nice-Sophia Antipolis, Centre Méditerranéen de Médecine Moléculaire (C3M), Department of microbial toxins in host-pathogen interactions, 151 Route St Antoine de Ginestière, BP 2 3194, 06204 Nice, France. [2] Immunology Program, Benaroya Research Institute at Virginia Mason, 1201 Ninth Avenue, Seattle, Washington 98101, USA. [3] LCP-A2MC, Institut Jean Barriol, Université de Lorraine, 1 bd Arago, Metz 57078, France. [4] Program in Cell and Molecular Biology, Institute of Biotechnology, P.O. Box 56, University of Helsinki, Helsinki 00014, Finland. [5] Department of Biology, University of Pennsylvania, 433 S. University Avenue, Philadelphia, Pennsylvania 19104, USA. [6] Multiscale Physics-Biology-Chemistry and Cancer, Cellular Microbiology and Physics of Infection Group, Center for Infection and Immunity of Lille, CNRS UMR8204, INSERM U1019, Institut Pasteur de Lille, Centre Hospitalier Régional de Lille, Université de Lille, Lille 59021, France. [7] Multiscale Physics-Biology-Chemistry and Cancer, Laboratoire Physico Chimie Curie, Institut Curie, PSL Research University, CNRS, UMR168, Paris 75005, France. [8] Multiscale Physics-Biology-Chemistry and Cancer, Sorbonne Universités, UPMC Univ Paris 06, Paris 75005, France. [9] Equipe labellisée La Ligue contre le Cancer. Correspondence and requests for materials should be addressed to E.L. (email: emmanuel.lemichez@inserm.fr) or to D.G.-R. (email: david.gr@univ-lorraine.fr) or to P.B. (email: patricia.bassereau@curie.fr).

The endothelium lining blood and lymphatic vessels forms a semi-permeable barrier separating body fluids from host tissues and is a major target of pathogenic bacteria[1–3]. Several bacterial toxins trigger the opening of transendothelial cell macroaperture (TEM) tunnels associated with vascular dysfunctions[4,5]. These tunnels show striking topological similarities with tunnels dug by immune or cancer cells in the endothelium and with intercellular gaps found in epithelial tissues during development and healing[6–10]. It is critical to define principles that govern the stabilization of gaps that form in the endothelium to control endothelial permeability[11]. Existing analogies with the physics of liquid dewetting provide a theoretical framework to better define biochemical and mechanical parameters underpinning the control of TEM size.

Several bacterial pathogens compromise the barrier function of endothelia by triggering the opening of transcellular tunnels as large as several micrometres in width[12]. The opening of TEMs occurs in response to the overall relaxation of the actomyosin cytoskeleton and cell spreading-associated with a disruption of focal adhesions via either toxin-induced inhibition of RhoA signalling or increase of the flux of cyclic-AMP[4,13]. This phenomenon has been linked to the dissemination of *Staphylococcus aureus* via a haematogenous route thereby promoting septic metastasis[14,15]. The opening of TEMs is transient, reaching a maximal size of $\sim 10\,\mu m$ in radius before resealing via membrane extensions[4]. The closure of TEMs consists of the extension of lamellipodia-like projections from the edges of TEMs via local Arp2/3-dependent actin polymerization driven by the I-BAR domain-containing protein MIM[4]. Depletion of MIM does not prevent actin recruitment along TEM edges and has a limited effect on their widening[5], thereby suggesting the existence of an additional process controlling TEM width.

The cell cortex forms a composite material made of a lipid bilayer tethered to cortical cytoskeleton components. It combines remarkable properties of strength and flexibility that are largely attributed to the actomyosin network. The concepts governing the physical and molecular relationship between both materials to adjust cell shape and cope with environmental stresses have been the subject of intensive research[16]. Different types of cytoskeletal molecules drive the local polymerization of F-actin and the tethering of filaments at the membrane, as well as the crosslinking of actin filaments into contractile actomyosin networks. For example, the recruitment of I-BAR domain-containing proteins, such as MIM and ABBA, at curved membranes generates protrusive forces due to the local polymerization of branched F-actin[4,17]. The ERM proteins ezrin, radixin and moesin are examples of critical linkers bridging cortical F-actin filaments to the plasma membrane in different types of microvilli[18,19]. Despite a high level of sequence identity among ERMs, this group of proteins displays non-overlapping functions[18–21]. ERM proteins are composed of an N-terminal FERM domain that is folded on the F-actin binding C-terminal region, which controls ERM auto-inhibition[22]. The phosphatidylinositol 4,5-bisphosphate (PIP$_2$) binding site in the FERM domain of ezrin is essential for its targeting from the cytosol to the plasma membrane, where a subsequent T567-phospho-dependent unmasking of the F-actin binding domain occurs[19,23,24]. Non-muscle myosin II (NMII) are principal active components of tensile and stiffness forces[16,25,26]. The NMIIa, b and c heavy chains of myosin II are encoded by three different genes that share the same regulatory light chains (myosin light chain, MLC). All three forms of NMII show differences in cell-type-specific expression and specific functions that still remain to be fully elucidated[26].

Here, we sought to decipher the mechanisms underlying the size control of TEM tunnels. We reveal the existence of two different actin structures along the edges of TEMs: (1) a previously described branched F-actin network forming lamellipodia-like structures extending towards the centre of the TEM; and (2) a bundle of actin filaments forming a circular cable around the TEM. We refine the cellular dewetting physical model to integrate the control of TEM size by line tension that is enhanced over time by the assembly of a stiff actomyosin cable. Mechanistically, we provide evidence that ezrin stabilizes F-actin at the edges of TEMs, thereby promoting the NMIIa-induced formation of this actomyosin cable.

## Results

**Formation of actomyosin cables along TEM edges.** The opening of TEM tunnels, as visualized in Supplementary Movie 1, has phenomenological similarities and differences with the principles governing liquid dewetting, as schematically represented in Fig. 1 and discussed in refs 5,12. Because of the phenomenological analogies between TEM opening and liquid dewetting, the former phenomenon has been termed cellular dewetting[5,12]. In contrast to spreading, dewetting refers to the nucleation and growth of dry patches when a viscous film is placed on a non-wettable surface. Upon nucleation of a dry patch, the surface tension in the liquid film generates forces that tend to enlarge the dry patch (Fig. 1a, black arrows). In liquids, the growth of dry patches is unlimited (Fig. 1a, top). In cells, the dry zone reaches a maximum size at which enlargement stops (Fig. 1a, bottom). This difference can be interpreted by the existence of a driving force for cellular dewetting that decreases as the size of the dry patch increases. This fading of surface tension is fairly local, given that other TEMs continue to form in the cell (Supplementary Movie 1). This favours the hypothesis that TEM size-dependent stabilization of forces corresponds to an increase of line tension at the TEM edges (Fig. 1a, green arrows). The molecular basis of line tension that a cell implements to maintain its physical boundaries remains to be established. Among possible hypotheses, line tension may arise because of the energy cost of bending a membrane that forms the ridge along the edges of TEMs. This hypothesis, together with a membrane tension that decreases with TEM size, were used in our earlier model of TEM dynamics to explain the equilibrium size of TEMs formed *de novo*[5]. A second, non-mutually exclusive hypothesis suggests the assembly of a rigid scaffold along the edges of TEMs. We mapped the local stiffness over the cell, as quantified by local values of the Young's moduli. The results revealed a line of high stiffness superimposed on actin staining along the edges of TEMs (Fig. 1b). We thus hypothesized that actin-dependent structures around the TEM might actively contribute to line tension. The value of the Young modulus around TEMs was in agreement with that reported for the elasticity of actin stress fibres analysed by atomic force microscopy (AFM) in living cells[27]. This prompted us to investigate the architecture of the actin cytoskeleton along the edges of the tunnels. Endothelial cells were intoxicated with exoC3 toxin to promote the formation of TEMs. The cells were then processed for electron microscopy analysis after detergent extraction under treatment conditions that stabilize actin structures[25]. As shown in Fig. 1d, we first observed that typical TEMs displayed at their edge a dense network of dendritic actin filaments, which underlie membrane ruffles. This network was similar to the previously observed actin structures driven by Arp2/3 in lamellipodia[28] and was in agreement with our former observation that MIM recruits and activates Arp2/3 at the edges of TEMs[4]. Interestingly, this approach also revealed the presence of an actin bundle encircling TEMs (Fig. 1d,e). In a complementary approach, we observed by immunolabelling a localization of myosin II around TEMs (Fig. 1e). Thus, two types

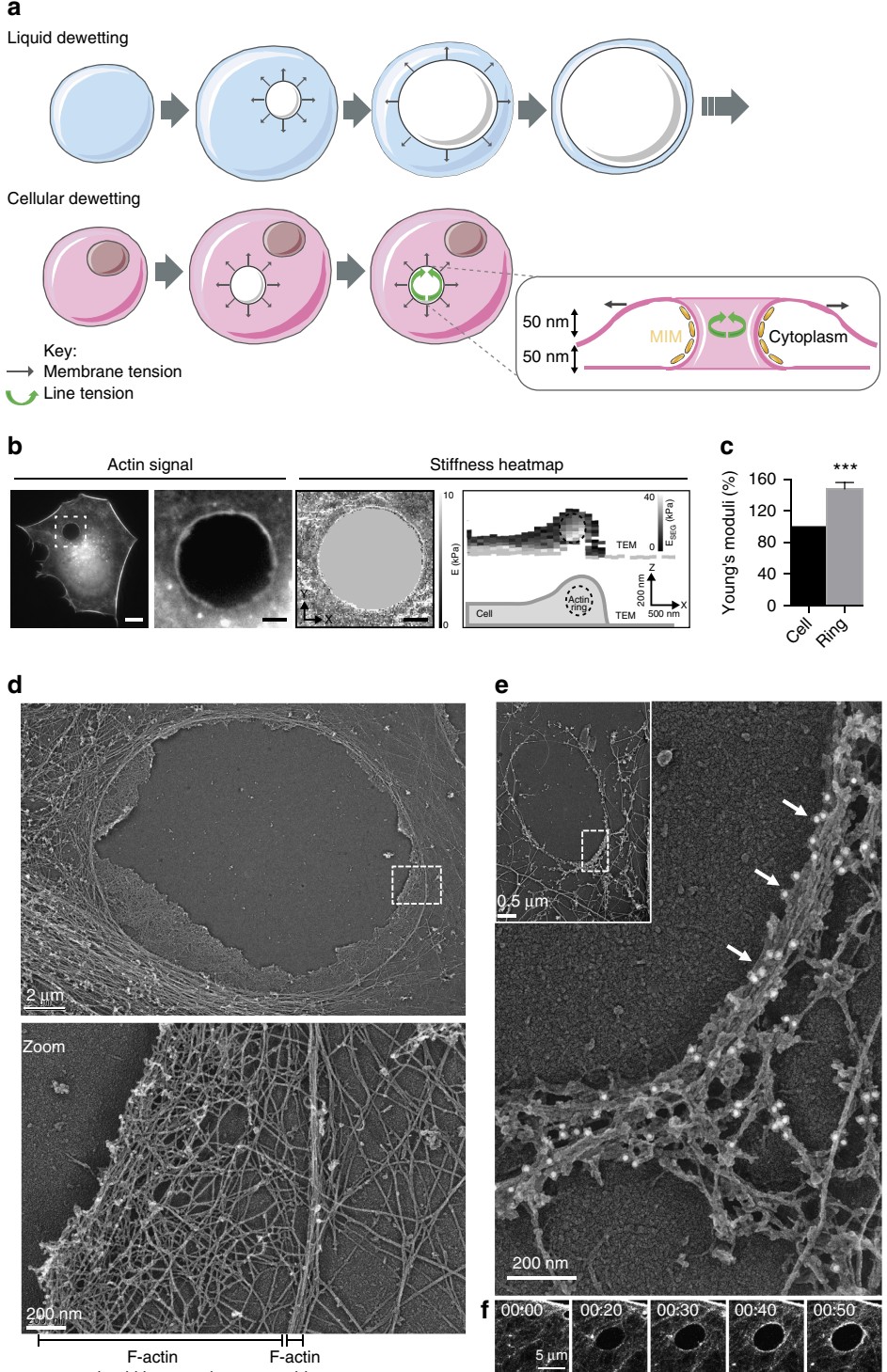

**Figure 1 | organization of the actomyosin cytoskeleton along TE edges.** (**a**) The dewetting phenomenon describes the rupture of a thin, viscous liquid film on a non-wettable surface, generating a dry patch that widens unlimitedly (top). In the cellular mode of dewetting, cells limit TEM enlargement to a diameter of approximately 10 μm. Holes widen as a function of the tension of the membrane that originates from cell spreading, exerting a tensile force on the tunnel (black arrows). Enlargement of the holes is resisted by a line tension along TEM edges (green arrows). (**b**) Elasticity of the actin ring encircling TEMs. Left: actin staining in a living cell. Centre: zoomed-in view of the TEM. Right: corresponding living cell elasticity map. Scale bars, white, 10 μm; black, 2 μm. Bottom: Elasticity tomogram showing the height profile together with the actin cable. (**c**) Bar plot of elasticity. Values correspond to the normalized medians of Young's moduli ± s.d. $n = 7$ TEMs. ***$P < 0.001$, student $t$-test. (**d**) Representative example of a platinum replica electron micrograph of TEMs in HUVECs after exoC3 treatment for 24 h. Zoom shows enlarged boxed region. This displays at higher magnification the organization of F-actin into a dendritic meshwork protruding into membrane waves and into an actin bundle at the rear. (**e**) Electron micrograph of gelsolin-treated cytoskeleton with NMII immunogold labelling at the edge of a TEM. Inset shows an overview of the TEM at lower magnification; boxed region correspond to the main panel. Scale bars are labelled. (**f**) Representative video images (Supplementary Movie 2) showing the accumulation of LifeAct-GFP signal around a TEM in HUVECs intoxicated for 24 h with exoC3. Scale bars are labelled.

of F-actin structures co-exist along the edges of TEMs: (1) a previously characterized branched network involved in lamellipodia-like membrane waves that invade the hole for closure (Supplementary Movie 1) and a yet uncharacterized actomyosin cable. Video analysis of cells expressing LifeAct-GFP revealed the accumulation of F-actin around TEMs during the opening phase (Fig. 1f and Supplementary Movie 2). This newly identified actin cable is a strong candidate structure to explain line tension development around TEM.

**A stiff actomyosin bundle encircles TEMs.** We assessed the mechanical stresses operating at the edges of TEMs. We released local forces acting along the edges of TEMs by sub-nanosecond laser severing of intracellular cytoskeletal structures. Since its development, laser ablation nanosurgery has proven to be a powerful method for analysing the mechanical properties of cytoskeletal structures[29–31]. Laser ablation was coupled to time-lapse video analysis of cells expressing LifeAct-GFP to monitor the actin cable dynamics. These experiments were performed on stable TEMs that, at the moment of the ablation, did not display membrane wave extensions engaged in the closure of the hole (Supplementary Movie 1). When we performed laser severing in the immediate vicinity of TEMs, instead of the edges of the tunnels, we observed no effect (Fig. 2a: control ablation and Supplementary Movie 3). In contrast, when ablation was performed at the edges of a TEM, we observed the widening of the tunnel (Fig. 2a: ablation at the edge and Supplementary Movie 3). Eventually, after a few seconds, the size of TEM stabilized, thereby reaching a second equilibrium. All tunnels enlarged in the direction of the cut, as schematized in Fig. 2a (directionality). The laser ablation technique allows quantification of the enlargement of TEM as a function of their initial size. Fitting our experimental data by comparing the TEM equilibrium radius before ($R_0$) and after ablation ($R_{eq}$) yielded $R_{eq} = R_0 + 2.0$ (μm), $n = 55$ (Fig. 2b). Strikingly, this result showed that the TEM radius changed by a constant amount independently of its initial value. When laser severing encompassed the entire TEM, we observed that holes underwent an isotropic widening (Fig. 2a: total ablation). We next studied the dynamics of the actin cytoskeleton after ablation. We observed that within the first 2 s of widening, the expanding zone was depleted of F-actin signal (Fig. 2c, graph). The actin cables encircling the TEMs maintained the same length before and just after laser severing (Fig. 2d). This result indicated that the actin cable resists TEM opening as a stiff inextensible structure rather than as an elastic deformable structure under tension. Next, as the hole grew, we detected a *de novo* accumulation of F-actin in this zone (Fig. 2c, black arrow). Analysis of LifeAct-GFP signal along enlarged TEMs allowed us to calculate a F-actin replenishment time in the expending zone (as shown in Fig. 2c, arrow) of 10.1 s ± 1.6 s.e.m., $n = 14$ TEMs, concomitantly with the TEMs reaching the new equilibrium radius, $R_{eq}$ (Fig. 2e). We verified that a second ablation produced a *de novo* enlargement of TEMs (Fig. 2f and Supplementary Movie 4). We thus showed that line tension at the TEM boundaries was evident in laser ablation experiments, thus highlighting the critical role of the actin cable structure.

**Physical model of TEM enlargement after laser ablation.** We went on to test mechanical hypotheses on the origin of line tension taking advantage of a previously developed cellular dewetting model describing *ex novo* TEM opening[5]. In that model, the dynamics of TEM opening are governed by the interplay between membrane tension, line tension and substrate friction. In support of a crucial role of plasma membrane mechanics in TEM opening, membrane perturbation by treating

cells with the detergent deoxycholic acid[32] reduced the mean area of TEMs by 3.77-fold (Supplementary Fig. 1). This physical picture is consistent with earlier biophysical models that have been proposed to interpret morphological changes in cells and tissues. For example, Sandre et al.[33] proposed a similar dewetting analogy to describe the opening of transient pores in a lipid membrane. Kozlov and Mogilner[34] explained cellular polarization and incipient motility through the interplay between actin-dependent line tension and membrane tension. Recently, Fedorov and Shemesh[35] published a physical model of *ex novo* TEM dynamics that accounts for membrane tension and line tension, in a fashion similar to our earlier model[5], coupled with an actin gel polymerization formalism to describe the dynamics of TEM closure. In a multicellular context, other biophysical models have described hole opening in cellular monolayers by an analogy with liquid dewetting[36], as well as hole closure in tissues, a phenomenon that shares certain physical ingredients with the model described here, such as the existence of an actin-dependent line tension and of a resisting force arising from friction with the substrate[37]. In the theoretical framework proposed in Gonzalez-Rodriguez et al., the net driving force per unit length of a TEM, $F_d$, is

$$F_d = 2\sigma - \frac{T}{R}, \qquad (1)$$

where $\sigma$ is the membrane tension, $T$ is the line tension and $R$ is the TEM radius. In principle, $\sigma$ depends on $R$, as described by Helfrich's law[38]. Nevertheless, in the setting of ablation experiments, we can approximate $\sigma$ by a constant value. Indeed, this approximation is justified because, if $R_0 < 10 \mu m$ (of the order of 1/5 of the cell radius of 50 μm), then $R$ can be considered small compared to the cell radius. Otherwise, if $R_0 > 10 \mu m$, the variation of $R$ after ablation ($\sim 2 \mu m$) can be considered small compared to $R$. Either of the two hypotheses allow us to approximate $\sigma$ by a constant. In the cellular dewetting model from ref. 5, the line tension, $T$, was attributed to the energy required to bend the membrane along the rim of the TEM. However, curvature-induced line tension should not be affected by laser ablation. Thus, this previous model is insufficient to explain our current observations that TEMs enlarge after ablation. Rather, the enlargement of TEMs after ablation indicates that the resisting force $T$ drops, and thus that line tension depends on the structural integrity of the actin cable. The ensuing TEM stabilization at a new maximum radius suggests that the resisting force can be rebuilt. Since TEM opening occurs at shorter timescales distinct from the beginning of TEM closure driven by lamellipodia-like membrane extensions, we can consider the two phenomena as temporally separated, and thus consider that the maximum TEM size corresponds to an equilibrium value. Using analytical modelling, we have investigated possible physical mechanisms underlying TEM stabilization upon ablation of the actin cable, as discussed in the Supplementary Note 1. Among these different hypotheses, we conclude that the only physical picture consistent with the experimental observations is that TEMs are stabilized by the reorganization of an actin cable around the rim, which eventually becomes strong enough to balance membrane tension. We thus suppose that line tension develops over the typical time required for the organization of the actin cable. This is the basis to derive an improved model, whose mathematical details are described in the methods section.

Figure 3a shows the improved model's prediction of the equilibrium radius after TEM ablation, $R_{eq}$, as a function of the initial radius before ablation, $R_0$. The predictions are in excellent agreement with the experimental measurements, because the model predicts an increase in the TEM radius after ablation that

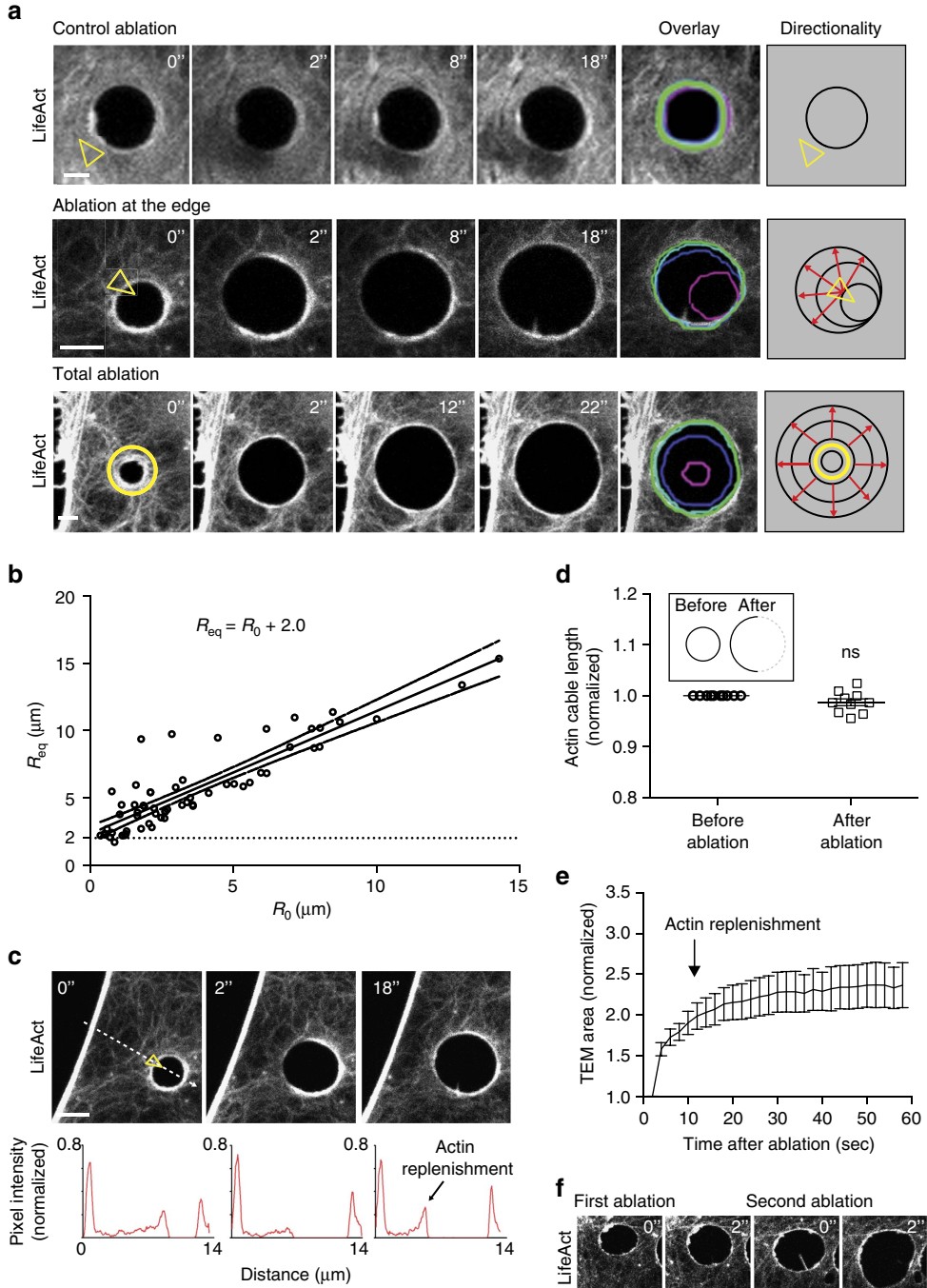

**Figure 2 | The actin ring prevents the widening of TEMs. (a)** Time-lapse video microscopy of TEM tunnels opening after laser ablation. HUVECs expressing LifeAct-GFP were intoxicated 24 h with exoC3. Yellow circles and arrowheads delineate the zone of ablation outside the TEMs, as a control for ablation (top), at the edges of the TEMs (middle), or around the entire TEMs (bottom). Left schematics show the enlarged zone and red arrows show the direction of enlargement. Scale bar, 5 μm. **(b)** Graph shows the linear relationship between the maximal values of radii before ($R_0$) and after ablation ($R_{eq}$), $n = 47$ TEMs. **(c)** Representative images showing the replenishment of F-actin in the depleted zone that enlarged following ablation. HUVECs expressing LifeAct-GFP were intoxicated for 24 hours with exoC3. Ablation is indicated by a yellow arrowhead. The graph beneath shows the value of the fluorescence signal intensity along the white arrow. Scale bar, 5 μm. **(d)** Graph shows the absence of variation of the length of the actin bundles along TEM edges before and after ablation ($n = 11$ TEMs). **(e)** Graph displays the mean values of the surface enlargement after ablation of TEMs as a function of time. The data are the means ± s.e.m., ($n = 3$, 58 TEMs of 5–10 μm of radius per condition). 'Actin replenishment' displays the timing at which the enlarged zone is replenished in F-actin signal over the background. **(f)** A second ablation produced a *de novo* enlargement of TEMs. Time-lapse video microscopy of TEM tunnel opening after laser ablation. HUVECs expressing LifeAct-GFP were intoxicated 24 h with exoC3. Scale bar, 5μm.

is virtually independent of $R_0$ and of ∼1.5 μm. The model also predicts the opening dynamics (equation (2) in the Methods section); it shows a reasonable agreement with the average opening dynamics of ablation experiments (Fig. 3b). Two model predictions are represented in Fig. 3b, the first one using the standard parameter values estimated in the Methods section

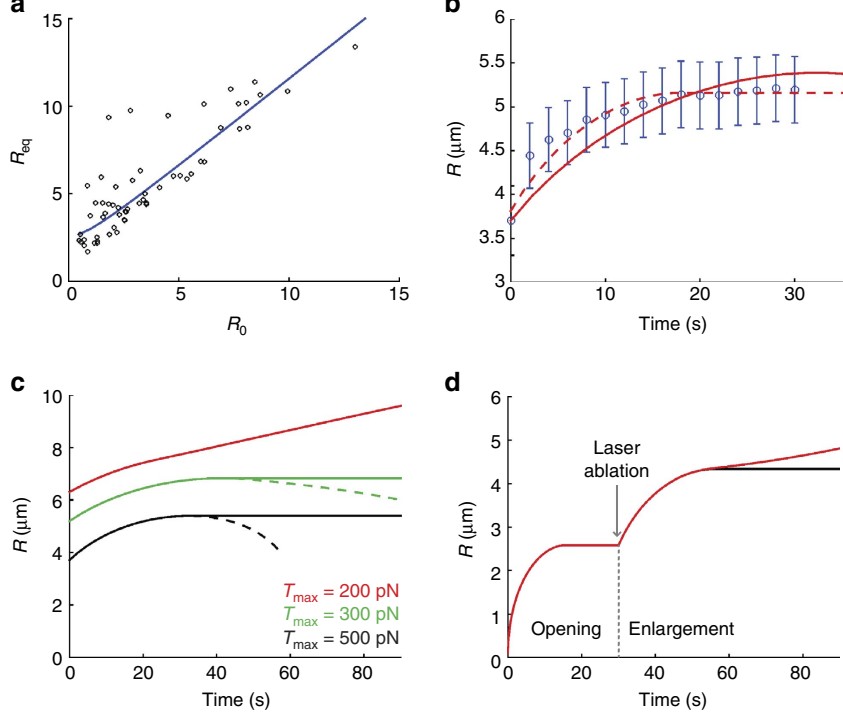

**Figure 3 | Physical interpretation of TEM enlargement after laser ablation.** (**a**) Comparison of the experimentally observed TEM opening after ablation and the model's predictions. The plot represents the final equilibrium radius, $R_{eq}$, versus the initial radius before ablation, $R_0$. The model is run using the standard parameter values discussed in the text ($\sigma = 2.5 \times 10^{-5}\,\mathrm{N\,m^{-1}}$, $\mu = 10^8\,\mathrm{Pa\,s\,m^{-1}}$, $\alpha = 8.3 \times 10^{-12}\,\mathrm{N\,s^{-1}}$). (**b**) Comparison of the experimentally observed dynamics of TEM opening under control conditions (blue symbols; mean ± s.e.m.) with the theoretical model's prediction, obtained using the standard parameter values of the physical parameters indicated in the text (solid red line) and using parameter values fitted to the data (dashed red line: $\sigma = 3.5 \times 10^{-5}\,\mathrm{N\,m^{-1}}$, $\mu = 10^8\,\mathrm{Pa\,s\,m^{-1}}$, $\alpha = 20 \times 10^{-12}\,\mathrm{N\,s^{-1}}$). (**c**) Effect of the maximum line tension, $T_{max}$, on TEM opening dynamics. The plot shows the TEM size increase as a function of time for the standard parameter values with $R_0 = 3.7\,\mu m$ (same as in plot **b**) and $T_{max} = 500\,\mathrm{pN}$ (black curve); for standard parameter values with $T_{max} = 300\,\mathrm{pN}$ and an initial $R_0$ increased by a factor of 1.4, representing siEzrin experiments (green curve); and for a case with $T_{max} = 200\,\mathrm{pN}$ and an initial $R_0$ increased by a factor of 1.7, representing siNMII experiments (red curve). Because the model does not describe the TEM closure mechanism, the predicted closure dynamics is represented by dashed lines and replaced by horizontal solid lines. (**d**) Schematic representation showing the model's prediction reproducing the two-phase TEM opening dynamics of an ablation experiment ($\sigma = 2.5 \times 10^{-5}\,\mathrm{N\,m^{-1}}$, $\mu = 10^8\,\mathrm{Pa\,s\,m^{-1}}$, $\alpha = 8.3 \times 10^{-12}\,\mathrm{N\,s^{-1}}$). The red curve corresponds to a reduced line tension, such as in siEzrin conditions ($T_{max} = 200\,\mathrm{pN}$). The curve shows a TEM opening *ex novo* (from an initial $R_0 = 100\,\mathrm{nm}$) and reaching a maximum equilibrium radius. If laser ablation is performed at the time indicated by the arrow, the TEM opens up again, without reaching a new equilibrium radius (red curve). In contrast, for a higher line tension ($T_{max} = 500\,\mathrm{pN}$, black curve), the TEM stabilizes again after laser ablation.

(solid red line), and the second, improved prediction, including parameter values that were adjusted to better fit the data (dashed red line).

Our advanced cellular dewetting model provides an accurate description of the experimentally observed equilibrium radii and opening dynamics, while allowing us to distinguish between mechanisms of line tension generation consistent with our data (actin cable polymerization, rearrangement or crosslinking) and those that are inconsistent (cable bending rigidity, transport of cable components to the rim by diffusion or by convection). Next, we develop the analysis further to predict how TEM key parameters affect TEM dynamics. These model predictions will next be compared to experiments conducted under specific cell treatments (Fig. 3c). Here, we are interested in investigating treatments that can impair the normal assembly of the actin cable. This can be modelled by considering that a fully mature cable can develop a maximum value of the line tension, which we denote $T_{max}$ (see Methods section for mathematical details). Indeed, our ablation experiments show that the cable is not contractile, since its length remains constant after ablation (Fig. 2d). TEMs close due to extension of lamellipodia-like structures, and not via a purse-ring cable-driven contraction, thus suggesting that line tension saturates at a maximum value, which can be estimated

from the largest observed radius of stabilized TEMs to be of about $T_{max} \approx 500\,\mathrm{pN}$ (see Methods). This estimate of typical line tension developed by the cable is consistent with typical tensions in single actin stress fibres, on the order of nanonewtons[39], and is much larger than the expected values of curvature-induced line tension, on the order of 5 pN (ref. 5). In Fig. 3c, we use the model to predict the effect of a treatment that would reduce the maximum tension the cable can develop (see also Supplementary Movie 5). The black curve corresponds to the standard parameter values chosen above and to an initial radius $R_0 = 3.7\,\mu m$, corresponding to the experimental data shown in Fig. 3b. The green curve corresponds to a reduction of the maximum tension that the actin cable can develop, from its standard value $T_{max} = 500\,\mathrm{pN}$ to $T_{max} = 300\,\mathrm{pN}$. Associated to the line tension reduction, a larger initial radius of the TEM was assumed because the equilibrium size of *ex novo* TEMs increases as the line tension decreases[5]. For both black and green curves, the TEM size stabilizes after laser ablation. The red curve shows that if $T_{max}$ is further reduced to 200 pN (while increasing the initial radius $R_0$ from 3.7 to 6.3 μm) TEM stabilization is eventually impeded and it keeps opening indefinitely. We have next investigated the effect of a reduced line tension depending on the TEM's initial size. Fig. 3d shows that with a reduced $T_{max} = 200\,\mathrm{pN}$ (red curve), a TEM formed anew

(at $t = 0$ on this graph) can still stabilize and reach an equilibrium size. However, if a laser ablation is performed (at the time indicated by the arrow), the TEM opens indefinitely. This difference between the two behaviours arises from the expression of the driving force, equation (1). If $T_{max}$ is not large enough, equation (1) may yield a zero net driving force for small TEMs (in the case of a TEM formed anew), but not for larger TEMs (in the case of the same TEM opening after ablation). In contrast, for a larger value of $T_{max}$, the TEM can stabilize after laser ablation, as illustrated by the black curve in Fig. 3d. We note that the red and black curves in Fig. 3d only differ in their value of $T_{max}$, whereas for simplicity α, the rate of line tension strengthening during cable assembly, is supposed to remain constant. Since $T_{max}$ is only reached after laser ablation, the two curves yield identical results over the first opening phase, for which $T < T_{max}$. Overall, we propose that TEM opening is resisted by line tension developed by the assembly of the actomyosin cable. According to our model's predictions, TEM stabilization at a maximum size is indicative of high line tension attained in the actomyosin cable, whereas a TEM that continues opening up indefinitely after laser ablation suggests a weaker cable that can develop only a smaller line tension. In the following, we have critically tested this model by perturbing cellular components expected to affect actin cable strength.

**NMIIa enhances line tension that resists TEM opening.** Since NMII has F-actin crosslinking function, we set out to test the effect of depletion of NMII heavy chains on TEM parameters. NMII depletion was verified (Supplementary Fig. 3). We directly visualized the effects of NMII depletion on cells displaying TEMs (Fig. 4a and Supplementary Movie 6) and performed quantitative measurements of TEM parameters (Fig. 4b,c). Video analysis provided evidence for greater heterogeneity in TEM size and the appearance of giant TEMs (Supplementary Movie 6). The depletion of NMII induced an increase in the percentage of cells displaying TEMs by 2.9-fold ± 0.3 s.e.m., $n > 50$ (Fig. 4b). The mean size of the TEMs increased by 1.7-fold ± 0.3 s.e.m., $n > 50$ (Fig. 4c). We then measured TEM area dynamics in NMII knockdown cells to obtain quantitative values of the intrinsic capacity of TEMs to enlarge immediately after the ablation (Fig. 4d and Supplementary Movie 7). We selected TEMs in siNMII-treated cells displaying typical initial radii of $R_0 = 8.7 \, \mu m$, as defined in Fig. 4c. Quantitative analysis of the variation of the radii over time after ablation revealed a critical implication of NMII in limiting their enlargement. Indeed, in NMII-knocked down cells, TEMs continued to grow after ablation at a constant speed over the study period of 350 s, while TEMs in control cells underwent closure (Fig. 4e). Pursuing the analysis further, revealed that 50% of TEMs were already closed at 500 s in control cells ($n = 6$), while 80% of TEMs in siNMII-treated cells continued to grow ($n = 6$). By comparing these observations with the theoretical model's predictions, depicted in Fig. 3c, we postulated that NMII limits TEM enlargement by enhancing line tension. In NMII-depleted cells, the maximum line tension developed in the cable decreased significantly, and thus the TEM size could no longer be stabilized, a result corresponding to the model's predictions for the red curve in Fig. 3c. Complementary to this quantitative approach, we detected an isoform-specific co-localization of endogenous and GFP-tagged NMIIa with the actin cables encircling TEMs (Fig. 4f). In contrast, no visible accumulation of NMIIb-GFP, around TEMs could be visualized (Fig. 4f). These results led us to assess the possible specific roles of NMIIa in TEM dynamics. We generated shNMIIa and shNMIIb expression vectors and verified their activity and specificity (Supplementary Fig. 4). In support of the results from the

localization studies, our isoform-specific knockdown experiments revealed that NMIIa is critical in limiting the percentage of cells displaying TEMs, whereas NMIIb KD had no effect (Fig. 4g). In conclusion, we revealed that NMIIa controls cell integrity by enhancing line tension along the edges of TEMs via a stiffening of the actin cable that resists TEM opening.

**Ezrin promotes actomyosin cable structuration at TEM edges.** Next, we carried out a candidate approach of critical ERM proteins known to bridge the plasma membrane to F-actin at curved membranes[19]. We performed RNAi-mediated knockdown of ezrin, radixin and moesin and then quantified the percentage of exoC3-treated cells displaying TEMs (Fig. 5a). The efficiencies of depletion were first verified by quantitative real-time PCR (Supplementary Fig. 5) and immunoblotting analysis for ezrin (Supplementary Fig. 6). Depletion of moesin or radixin had no significant effect, in contrast to the depletion of ezrin, which produced an increase in the percentage of cells displaying TEMs (1.81-fold ± 0.07 s.e.m., $n = 7$) (Fig. 5a). In these conditions of ezrin depletion, TEM displayed a high heterogeneity of size, with the formation of giant TEMs (Fig. 5b and Supplementary Movie 8). We measured a 1.36-fold ± 0.2 s.e.m. increase in the surface area of TEMs (Fig. 5c). We then performed laser ablation experiments on TEMs of size $R_o = 7.63 \, \mu m$ corresponding to mean value in condition of ezrin KD. This revealed a dramatic increase of TEM widening after ablation in ezrin KD cells (Fig. 5d and Supplementary Movie 9). Quantitative analysis revealed that TEMs in siEzrin KD cells enlarged during the first 240 s after ablation before reaching a plateau, compared with 40 s in control conditions, thus leading to larger TEMs (Fig. 5e). It is interesting to draw a parallel between these results and our electron microscopy data showing thinner cables along the edge of TEMs that form in ezrin-depleted cells compared to control cells (Fig. 5f,g). This suggested that the silencing of ezrin compromised TEM stabilization by impairing the implementation of line tension in the actin cable. Indeed, comparison of the observed TEM opening dynamics in Fig. 5e with the theoretical model's predictions in Fig. 3c (green curve) suggests that silencing ezrin reduces the line tension in the actomyosin cable, although this reduction is smaller than that due to NMII depletion. The line tension in siEzrin experiments was barely high enough to balance membrane tension and stabilize the TEM. Supporting this conclusion, the number of TEMs that reached an equilibrium size was comparable to the number of those that continued to open indefinitely. We thus conclude that ezrin contributes to stabilizing TEM width by orchestrating the organization of an actomyosin cable along TEM edges that accounts for the increase of line tension.

**Ezrin role in the formation of actin bundles along TEM edges.** We then further studied the functional relationship between ezrin and TEMs. Confocal analysis revealed an accumulation of endogenous ezrin and ezrin-GFP signal along TEM edges (Fig. 6a). The open-conformation mutant T567D form of ezrin also showed a marked accumulation around TEMs, where it co-localized with F-actin (Fig. 6a). In parallel, our quantitative analysis showed that the expression of ezrin-T567D-GFP reduced the percentage of cells displaying TEMs to 24.1% (± 0.3% s.e.m., $n > 50$ TEMs) and decreased the mean size of these TEMs by 0.78-fold (± 0.01% s.e.m., $n > 50$ TEMs) (Fig. 6b,c). Moreover, we linked the effect of ezrin-T567D to its activated state, because the expression of the wild-type form of ezrin in intoxicated cells showed no significant effect (Fig. 6b,c). Together, our data suggested a hierarchy of accumulation between ezrin, F-actin and NMII. We returned to laser ablation experimental settings to

assess this hypothesis, using LifeAct-GFP as a standard. We measured that ezrin-GFP accumulated at the edges of TEMs 8 s after ablation, concomitantly with the accumulation of LifeAct-GFP (Fig. 6d). We measured a belated recruitment of MLC-GFP 24 s after ablation, that is, 16 s after that of ezrin (Fig. 6d). Thus, the accumulation of ezrin and F-actin at enlarged

TEMs precedes the accumulation of MLC. From these data, we hypothesized that ezrin might stabilize F-actin along the edges of TEMs. In support of this hypothesis, we have observed in electron micrographs of ezrin-depleted cells that F-actin bundles at the edges of TEMs were loose and thin (Fig. 5f). Finally, we no longer observed the significant accumulation of NMIIa along the TEM

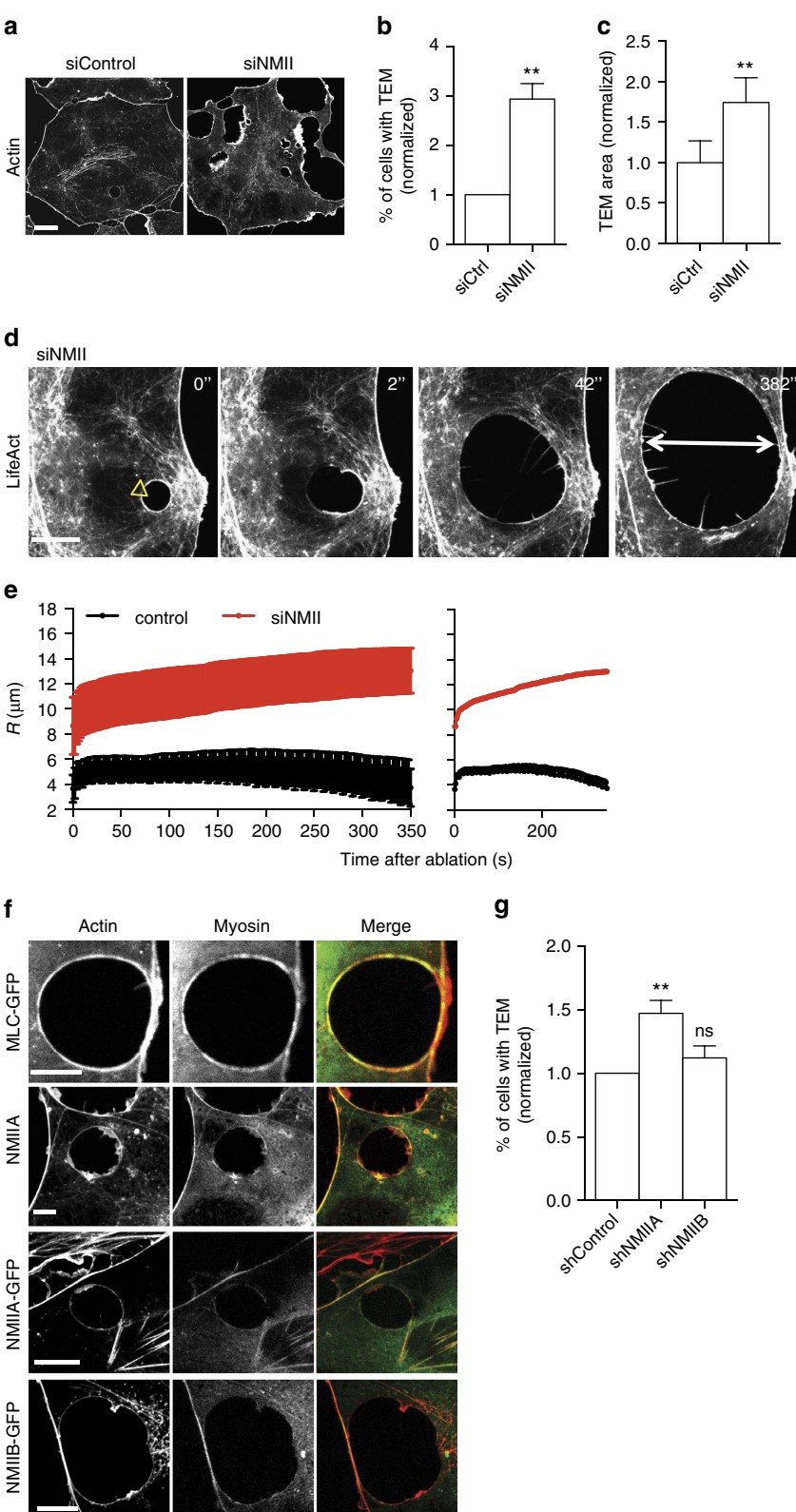

in conditions of ezrin depletion (Supplementary Fig. 7a). To challenge our hypothesis that ezrin might contribute to stabilize F-actin around TEMs, we performed a fluorescence recovery after photobleaching (FRAP) analysis of the GFP-actin signal in control and ezrin KD cells. In these FRAP experiments, we bleached the entire circumference of the TEMs to ignore a possible contribution of the lateral dynamics of F-actin in the recovery process. The FRAP analysis revealed that the depletion of ezrin leads to a 1.3-fold increase in the recovery rate of F-actin (Ctrl $t_{1/2} = 4.6$ s versus siEzrin $t_{1/2} = 3.6$ s) and to a 1.4-fold increase of the mobile fraction (Fig. 6e). Hence, we conclude that ezrin stabilizes F-actin along the edges of TEMs, thereby promoting the bundling of F-actin and the reinforcement of line tension.

## Discussion

By combining laser ablation experiments with a physical model derived by analogy with dewetting theory, we provide compelling evidence that TEM size is restricted by the organization of an actomyosin belt encircling TEMs. This belt develops line tension over time, thereby dictating TEM maximal size. Mechanistically, we propose that ezrin, by stabilizing actin filaments around TEMs, favors the NMIIa-dependent bundling of F-actin into a stiff cable. Our results indicate that ezrin contributes to cable stiffness by increasing its line tension, thus affecting TEM maximum size. This ascribes to ezrin molecule a function of control of endothelial cell integrity.

It is of great interest to define the bidirectional interplay between membrane architecture and the dynamic assembly of actin cytoskeleton structures. Much remains to be learned about how actin cables are built at retracting cell boundaries, which contrasts with the significant progress made in defining how they form at membrane protrusions[16,40–42]. The extent of myosin-based contractility controls the switch between branched and bundled actin-based structures that plays a critical role in tissue organization versus cell polarization and motility[43]. Formation of actomyosin cables along TEMs occurs in a peculiar cellular background of low RhoA-driven myosin contractility and membrane retraction. Thus, the low myosin contractility favours the formation of dendritic actin network, which is likely compensated by a high compression along TEMs due membrane receding that is responsible for a local increase of F-actin density and bundling. This thereby allows both actin structures to co-exist in condition of low RhoA activity. Actomyosin rings are critical cytoskeletal structures contributing to essential processes, such as cytokinesis and wound healing, in epithelia[9,44–47]. Actomyosin ring is also critical to adjust the size of tunnels between endothelial cells when neutrophils undergo endothelial diapedesis[8]. In contrast, little is known on mechanisms controlling the size of tunnels in single cells. We have previously reported that the recruitment of MIM

for lamellipodia-like formation at the edges of TEMs is under the control of its I-BAR domain, which contains positively charged amino acids and an amino-terminal amphipathic α-helix[4,17]. Here, we found that ezrin and its FERM domain accumulates around TEMs (Supplementary Fig. 7b). The precise mechanism that underlies this accumulation needs to be established. This may be linked to an enrichment of phosphatidylinositol-(4,5) $P_2$ (PIP$_2$) visualized by accumulation of the GFP-tagged probe of the pleckstrin homology (PH)-domain from the phospholipase Cδ (PH-PLCδ-GFP) (Supplementary Fig. 6c). Moreover, we here defined the hierarchy of the recruitment of ezrin and F-actin before the accumulation of NMIIa, consistent with ezrin promoting the local stabilization of F-actin. This points to other possibilities. One hypothesis is that ezrin forms of a diffusion barrier for actin filaments that would otherwise flow over the edges of TEMs during enlargement. A second non-mutually exclusive hypothesis is that ezrin stabilizes the half-life of F-actin that forms around TEMs. Although the mDIA1 formin displays a mechanosensitive actin nucleation activity when cell tension is released[48], we did not detect its accumulation at TEM edges. This second hypothesis would then indicate the role of a yet uncharacterized actin nucleation factor along TEM edges. In both hypothesizes, it results in the accumulation of actin filaments, which facilitates their subsequent bundling by NMII, thereby providing an upstream mechanism of actin bundle formation promoted by local enrichment of ezrin. The specificity of accumulation of NMIIa versus NMIIb cannot be explained solely by the accumulation of F-actin. A previous study has established the temporal hierarchy of the recruitment of NMIIa followed by NMIIb in cables, which is attributed to differences in their exchange with F-actin[25]. Another hypothesis might implicate a feedback mechanism in which membrane curvature stabilizes NMIIa cortical association[49]. Collectively, these findings indicate that several geometrical and biochemical spatio-temporal determinants likely contribute to the sorting of ezrin and NMIIa along the edges of TEMs.

The cellular dewetting model offers a theoretical framework to define mechanical parameters at play in membrane dynamics. The experimental observations presented here were interpreted through a dewetting model, accounting for a line tension that increases linearly over time upon actin cable assembly at a constant rate α, up to $T_{max}$, the maximum line tension developed by the fully mature cable. We interpreted the NMII and ezrin depletion conditions by a reduction of the parameter $T_{max}$, whereas for the sake of simplicity, we assumed that the value of α remained unchanged. In reality, α may also depend on NMIIa and ezrin, although this additional effect would not affect the conclusions reported here. We propose that in NMII-depleted cells, and in ezrin-depleted cells to a lesser extent, the maximum line tension developed in the cable decreases significantly. Indeed, physical modelling provides a rationale for this behaviour, since

**Figure 4 | NMIIa controls TEM size.** (**a**) Effect of NMII knockdown on TEM size and morphology. Cells were treated 24 h with NMII RNAi before exoC3 treatment for 24 h. F-actin was labelled with FITC-phalloidin. Scale bar, 10 μm. (**b**) Fold variations of the percentage of cells displaying at least one TEM. (**c**) Fold variations of the TEM area. Control conditions were set to a value of 1. HUVECs were transfected for 24 h with RNAi control (siCtrl) or RNAi targeting the different forms of NMII (siNMII). The cells were then treated for 24 h with exoC3. (**b,c**) The data are the means ± s.e.m. ($n = 3,400$ cells per condition). **$P < 0.01$, **b** is student $t$-test, **c** is Mann–Whitney. (**d**) Widening of the TEMs induced by laser ablation (yellow arrowhead) in HUVECs treated with NMII RNAi. Time-lapse video microscopy in LifeAct-GFP-expressing HUVECs treated for 24 h with siNMII and for 24 h with exoC3. Time in seconds after ablation. Scale bar, 10 μm. (**e**) Graphs showing the variation of the radius over time after ablation in HUVEC treated 24 h with siRNA control (siCtrl) or siRNA NMII (siNMII) and treated 24 h with exoC3. Data are represented as means ± s.e.m. (left) and only the mean curve (right). ($n = 3$ ctr and $n > 5$ siNMII). (**f**) Immunofluorescence analysis showing the localization of NMIIa-GFP (NMIIA-GFP and NMIIA endogenous) and the NMII light chain (MLC-GFP), in contrast to NMIIB-GFP, along the actin bundle encircling the TEM. Scale bars, 10 μm. (**g**) HUVECs were transfected for 24 h with shRNA control (shCtrl) or shRNA targeting NMIIa (shNMIIa) or NMIIb (shNMIIb). Fold variations of the percentage of cells with TEMs in cells expressing shNMIIA or NMIIB, compared with cells under control conditions (shControl), which were set to a value of 1. The data are the means ± s.e.m, ($n = 3,400$ cells per condition). **$P < 0.01$, ANOVA with Bonferroni $post$-$hoc$ test.

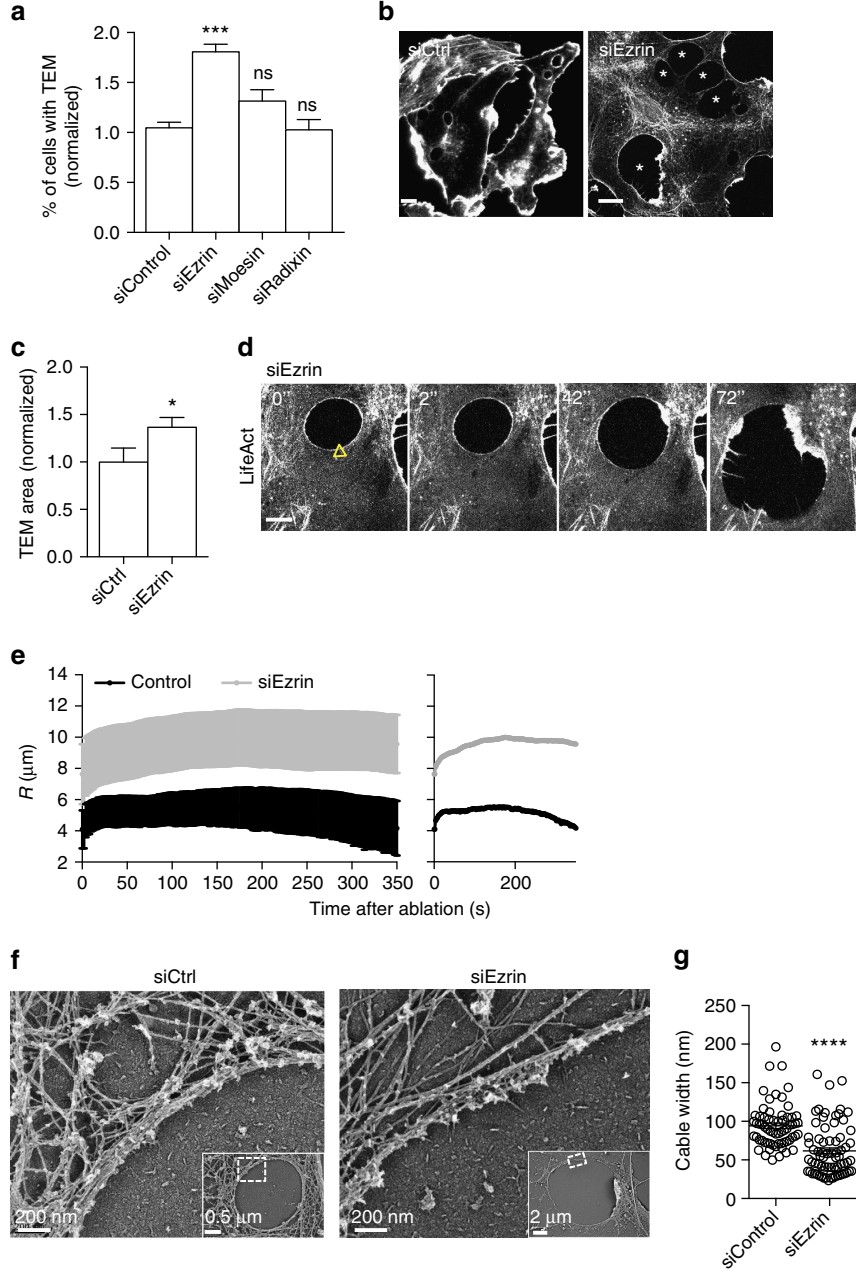

**Figure 5 | Ezrin controls TEM size.** (**a**) Percentage of cells displaying TEMs in HUVECs treated with control RNAi or RNAi targeting ezrin, moesin or radixin, compared with cells under control conditions (siControl), which are set to a value of 1. The data are the means ± s.e.m., ($n = 3,400$ cells per condition); ***$P < 0.001$, ANOVA with Bonferroni *post-hoc* test. HUVECs were transfected for 24 h with siRNA before cell treatment with exoC3 for 24 h. (**b**) Visualization of the effects of siEzrin on TEM size in HUVECs labelled with FITC-phalloidin. Giant TEMs are indicated with stars. The cells were treated for 24 h with ezrin RNAi before 24 h of treatment with exoC3. Scale bar, 10 μm. (**c**) Fold variations of the TEM area in siEzrin-treated cells compared with cells under control conditions, which were set to a value of 1. HUVECs were transfected for 24 h with siRNA control (siCtrl) or siRNA ezrin (siEzrin) before treatment with exoC3 for 24 h. *$P < 0.05$, student *t*-test. (**d**) Widening of TEM induced by laser ablation (yellow arrowhead) in HUVECs treated with ezrin siRNA. Time-lapse video microscopy in LifeAct-GFP-expressing HUVECs treated 24 h with siEzrin and 24 h with exoC3. Time in seconds after ablation (images from Supplementary Movie 9). Scale bar, 5 μm. (**e**) Graphs showing the variation of the radius over time after ablation in HUVECs treated for 24 h with siRNA control (control) or siRNA ezrin (siEzrin) and treated for 24 h with exoC3. Data are represented as means ± s.e.m. (left) and only the mean curve (right), $n > 5$. (**f**) Representative examples of platinum replica electron micrographs of TEM's edges in HUVECs treated for 24 h with siRNA control (siCtrl) or siRNA ezrin (siEzrin) and then treated for 24 h with exoC3. Insets show overviews of the TEMs at lower magnification; boxed regions in insets correspond to main panels. Scale bar, 200 nm. (**g**) Graph shows quantifications of the width of the cytoskeleton cable along the edge of TEMs in siControl treated cells (64 measures, $n = 2$ TEMs), compared to siEzrin-treated cells (72 measures, $n = 3$ TEMs). Two-tailed *t*-test, *** $P < 0.0001$.

the model shows that a moderate decrease in line tension may still allow the stabilization of *ex novo* TEMs, while impairing the stabilization of TEMs reopening upon laser ablation. Indeed, due to geometrical effects arising from the circular shape of TEMs, the stabilizing force generated by line tension is inversely proportional to TEM radius, as described by the last term in equation (1). Thus, a given magnitude of line tension $T$ will result in a higher stabilizing force in smaller, *ex novo* TEMs than in

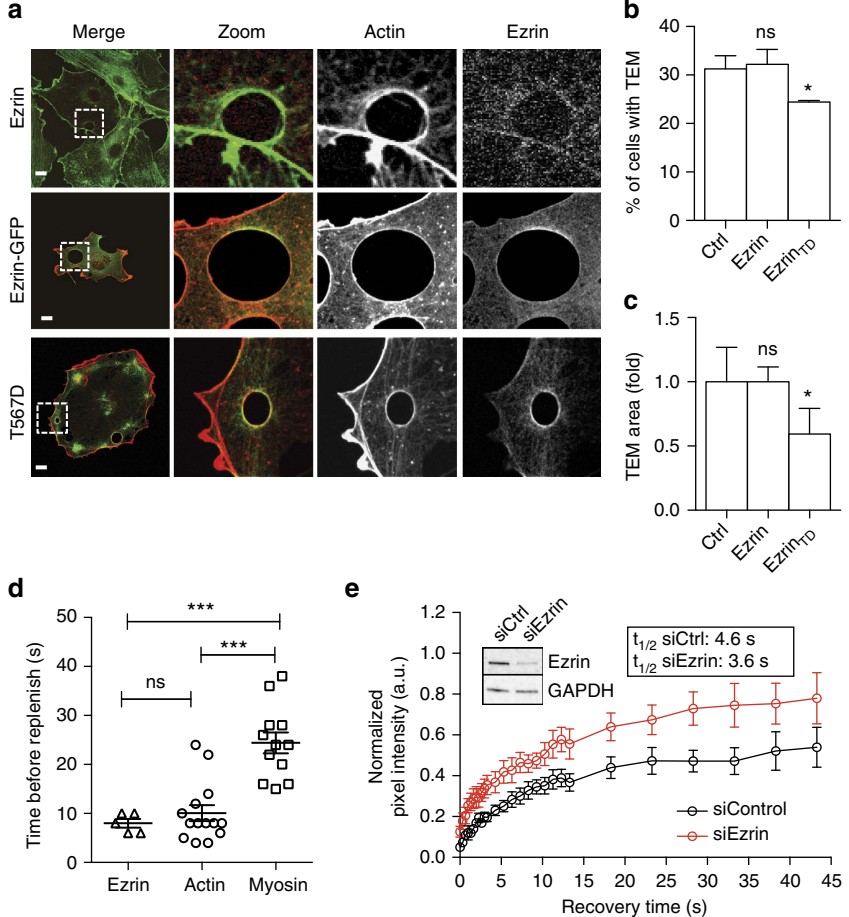

**Figure 6 | Ezrin orchestrates the formation of actin bundles along TEM edges.** (**a**) Localization of ezrin and ezrin-GFP constructs at the edges of TEMs. HUVECs were transfected with plasmids encoding ezrin-GFP, ezrin-FERM-GFP or the phosphomimetic ezrin-(T567D). Cells were next treated for 24 h with exoC3. Actin was labelled with phalloidin (red). Scale bar, 10 μm. (**b,c**) HUVECs were transfected for 24 h with plasmids encoding GFP control (GFP), ezrin (Ezrin) or the open/active form of ezrin (Ezrin-TD) before cell treatment for 24 h with exoC3. The data are the means ± s.e.m., ($n = 3,400$ cells per condition). NS, not significant; *$P < 0.05$, ANOVA with Bonferroni *post-hoc* test. (**b**) Graph show fold variations of the percentage of cells with TEMs compared to control conditions, which are set to 1. (**c**) Fold variations of the TEM area compared to control conditions, which are set to 1. (**d**) Hierarchy of recruitment after ablation of ezrin, LifeAct (Actin) and MLC-GFP (Myosin) to enlarged TEM zone. Each measure is represented together with means ± s.e.m., ($n \geqslant 5$ cells per condition). NS, not significant; ***$P < 0.001$, ANOVA with Bonferroni *post-hoc* test. (**e**) Fluorescence recovery after photobleaching (FRAP) analysis of actin turnover in actin cables at the TEM edge. FRAP was performed in EGFP-actin-overexpressing wild-type and ezrin knockdown cells. The graph shows the fluorescence recovery of the bleached region in control (black plots) and ezrin knockdown (red plots) HUVECs. The pre-bleaching value was normalized to 1. Mean values of half time = 4.6 (s) for control RNAi and $t_{1/2} = 3.6$ (s) for ezrin RNAi. Mean values of mobile fraction = 0.484 for control RNAi conditions versus = 0.680 for siEzrin. The data represent the mean values ± s.e.m. $n = 10$ for each group.

larger TEMs reopening after laser ablation. Resistance to compression of the cytoskeletal network is another additional physical phenomenon that can contribute to TEM stabilization. Assuming a linear elastic behaviour characterized by an elastic constant, $k$, the cytoskeletal resistance to compression can be represented by a term of the form $-k(R - R_0)$, to be added to the right-hand side of equations (1) and (2) (see Methods for equation (2)). Mathematically, this term has the same time dependence as the line tension term. Therefore a role for cytoskeletal resistance in TEM dynamics is compatible with the experimental observations under control conditions. However, cytoskeletal compression alone predicts an equilibrium TEM size. Thus, cytoskeletal compression does not suffice to explain the long-term dynamics after ablation observed under ezrin and NMII depletion, where TEMs may continue to open up indefinitely without reaching an equilibrium size. Therefore, whereas resistance due to compression of the cytoskeletal network may play a role in slowing down TEM dynamics, we conclude that the dominant resistance mechanism

to TEM opening is the line tension produced by the actomyosin cable forming around TEMs.

Actomyosin rings are critical cytoskeletal structures contributing to adjust endothelium permeability[1–3,8]. RhoA-dependent contractile actomyosin cables in adjacent cells adjust the size of the holes in the endothelium by which neutrophils pass[8]. Confining the size of the tunnels prevents vessel leakage. These data are in good agreement with our findings that a complete inhibition of RhoA by means of vectorized RhoA-inhibiting toxin leads to unlimited enlargement of TEM tunnels associated with vascular leakage in mice[11]. Here, in moderate conditions of inhibition of RhoA we characterized the actomyosin ring resisting the opening of TEMs as a stiff, inextensible structure. The absence of TEM closure by a purse-ring actomyosin contraction likely reflects a combination of low RhoA activity condition and the large size of TEMs.

Infectious processes largely rely on the capacity of microbes to corrupt actin cytoskeleton organization and promote large-scale deformations of cellular membranes[50–52]. ERM proteins, notably

ezrin and moesin, are critical determinants involved in the deformation of membranes during infection, as occurs during the invasion of epithelial cells by Shigella[53] or endothelium colonization by Neisseria meningitidis[54]. Here, we ascribe to ezrin an endothelial protective function of limiting the transcellular tunnel width in response to a TEM tunnel-forming toxin via the structuration of an actomyosin cable at cell boundaries.

## Methods

**Mathematical model.** We modify our previous model of cell dewetting[5] by postulating that line tension is described by a function of time, $T = T(t)$. We note that this adjustment of the dewetting model, where $T$ is no longer a constant but a function of time, does not affect the good agreement between the model and the experimentally observed dynamics of the opening of TEMs reported previously[5]. Indeed, because the membrane tension component of the driving force for opening is unchanged, the dynamics of ex novo TEM opening over a short time remain described by an equation of the form $R \sim t^{1/2}$, as discussed in ref. 5. The dynamics of TEM opening are described by

$$\mu R \frac{dR}{dt} = 2\sigma - \frac{T(t)}{R}, \quad (2)$$

where $\mu$ represents the viscous friction of the cell gliding over the substrate. We investigate analytically which form of the function $T(t)$, if any, yields results that are consistent with experimental observations. First, we consider the limit of TEMs whose size increase after ablation, $\Delta R = R - R_0$, is small compared to their initial size, $R_0$, that is, $R = R_0(1 + \varepsilon)$ with $\varepsilon \ll 1$. By linearizing the differential equation (2), we obtain

$$R_{eq} = R_0 + \Delta R = R_0 + \frac{1}{\mu} \left[ \frac{2\sigma t_{eq}}{R_0} - \frac{\int_0^{t_{eq}} T(t)dt}{R_0^2} \right], \quad (3)$$

where $R_{eq}$ is the maximum equilibrium radius attained at time $t_{eq}$. For equation (3) to be consistent with our observations that $\Delta R = $ Constant, the dependence on $R_0$ of the term in square brackets needs to vanish. This is achieved if $t_{eq} = R_0/v$, where $v$ is a constant with units of velocity, and $T(t) = \alpha t$, where $\alpha$ is a constant with units of force per unit time, representing the speed of strengthening of the actin cable during its assembly. The general dynamics of TEM opening are thus described by equation (2) with $T = \alpha t$, indicating that cable strengthening proceeds at a constant rate rather than being a diffusion-dominated process. To obtain dimensional predictions, numerical values must be assigned to the parameters. The membrane tension in our highly spread cells has been estimated to be on the order of $\sigma \approx 2.5 \times 10^{-5}\, N\, m^{-1}$ (refs 32,55). The friction coefficient has been estimated to be $\mu \approx 10^8\, Pa\, s\, m^{-1}$ (ref. 5). To reach an equilibrium radius, line tension must eventually balance membrane tension and yield a zero net driving force (equation (1)). Experimentally, we observed that an equilibrium radius of approximately $5\,\mu m$ was attained approximately $30\, s$ after ablation. We thus estimate $\alpha = 2\sigma R_{eq}/t_{eq} \approx 8 \times 10^{-12}\, N\, s^{-1}$. The model predicts an increase in the TEM radius after ablation that is virtually independent of $R_0$ and of $\Delta R \approx 2\sigma^2/(\alpha\mu) \approx 1.5\,\mu m$.

Next, we study how the model's parameters affect TEM dynamics. Specifically, we are interested in investigating treatments that can impair the normal assembly of the actin cable. This can be modelled by considering that a fully mature cable can develop a maximum value of the line tension, which we denote $T_{max}$. Line tension evolution due to actin cable assembly is thus described by

$$T(t) = \begin{cases} \alpha t & \text{if } t < T_{max}/\alpha, \\ T_{max} & \text{otherwise.} \end{cases} \quad (4)$$

Here $t$ is the time elapsed after laser ablation. Because TEMs as large as $10\,\mu m$ could be stabilized, we estimate that in control conditions $T_{max} \geqslant 2\sigma R_{eq} \approx 500\, pN$. As indicated by equation (4), the maximum line tension is reached over a time $T_{max}/\alpha$ which, in control conditions, is larger than $t_{eq}$ ($\sim 60$ versus $\sim 30\, s$). This allowed us to disregard $T_{max}$ in our initial analysis of control conditions.

**Numerical analysis.** The general dynamics of TEM opening, not restricted to the particular limit where $\Delta R \ll R_0$, are described by equation (2) with $T(t)$ given by equation (4). To numerically solve equation (2), it is useful to rewrite the equation in non-dimensional terms:

$$\frac{d\hat{r}}{d\hat{t}} = \frac{2(1 + \rho\hat{r} - \hat{t})}{(1 + \rho\hat{r})^2}, \quad (5)$$

with the initial condition $\hat{r}(0) = 0$. Here $\hat{r} \equiv (R - R_0)\alpha\mu/(2\sigma^2)$ is the non-dimensional radius increase, $\hat{t} \equiv \alpha t/(\sigma R_0)$ is the non-dimensional time, and $\rho \equiv 2\sigma^2/(\alpha\mu R_0)$ is a parameter measuring the radius increase with respect to the initial radius, $\rho \sim \Delta R/R_0$. The numerical solution of equation (5) is performed using Matlab.

**Cell culture and transfection.** Human umbilical vein endothelial cells (HUVECs) (PromoCell GmbH, Heidelberg, Germany) were grown and used between passages two and four. Cells were transfected, as described in (ref. 56). Intoxications were performed with recombinant purified exoC3 toxin at $100\,\mu g\, ml^{-1}$, as described in ref. 13. Plasmids expressing shRNA targeting MYH9 and MYH10 were generated as described in ref. 13 targeting sequences 5′-GGCGACCTATGAGCGGATG-3′ and 5′- ACAAACTCCAGGCCCTTGG-3′ respectively. SiRNAs were transfected using PolyMag reagent (OZ Biosciences, Marseille, France) following the manufacturer's procedure. RNAi targeting ezrin, NMII, Moesin, Radixin were purchased (Dhamacon, ThermoFisherScientific, Inc, St Leon-Rot, Germany).

**Reagents.** We used plasmid encoding the following constructs: LifeAct-mCherry (gift from P. Chavrier, Institut Curie, Paris); LifeAct-GFP (Ibidi, GmbH, Planegg/Martinsried, Germany); ezrin constructs encompassing ezrin-GFP, ezrin-FERM-GFP, ezrin-T567D-GFP (gift from M. Arpin, Institut Curie, Paris); NMII encoding constructs: NMIIa-GFP and NMIIb-GFP (Addgene, Cambridge, MA, USA); pEGFP-actin[41] and C3 exoenzyme encoding plasmid pEF-C3 (ref. 13).

For immunofluorescence and immunoblotting experiments, we used antibodies against ezrin (Santa Cruz, Biotechnology, Inc, Heidelberg, Germany, #sc-58758, dilution 1/1,000 for IB and 1/50 for IF), NMIIa (Santa Cruz, MYH9: [clone H-40], #sc-98978, dilution 1/1,000 for IB and 1/100 for IF), NMIIb (MYH10:[clone A-5], Santa Cruz, #sc-376954, dilution 1/1,000 for IB and 1/100 for IF), and mouse monoclonal antibody against beta-actin ([clone AC-74], Sigma, St Louis, MO, USA, #A5316, 1/10,000). Full western blot scans are shown in Supplementary Figure 8.

**Immunofluorescence.** Immunofluorecence analyses were performed on cells fixed in 4% paraformaldehyde (Sigma, St Louis, MO, USA). Actin cytoskeleton was labelled using $1\,\mu g\, ml^{-1}$ fluorescein isothiocyanate (FITC) (Sigma #P5282) or tetramethylrhodamine (TRITC)-conjugated phalloidin (#P1951, Sigma). Rabbit and mouse antibodies were detected using the following secondary antibodies purchased from Thermo Fisher: goat anti-rabbit Alexa Fluor488 (#A11034, dilution 1:500), goat anti-rabbit Alexa Fluor546 (# A11010, dilution 1:500), goat anti-mouse Alexa Fluor488 (# A11001, dilution 1:500) or goat anti-mouse Alexa Fluor546, (#A11003, dilution 1:500). Fluorescent signals were analysed with an LSM510-Meta confocal microscope using $\times 63$ or $\times 25$ magnification lens (Carl Zeiss, Göttingen, Germany). To avoid quantification bias, the microscopy examination was performed in a blinded manner.

**Immunoblotting.** For immunoblotting proteins were resolved on 12% SDS–polyacrlamide gel eletrophosis followed by transfer on polyvinylidene difluoride membrane (Millipore, Merck, KGaA, Darmstadt, Germany). Immunoblotting was performed using primary antibodies revealed with secondary goat anti-mouse (#P0447, dilution 1:5,000) or anti-rabbit (#P0399, dilution 1:5,000) horseradish peroxidase-conjugated secondary antibodies (DAKO, Glostrup, Denmark) followed by chemiluminescence using Immobilon Western (Millipore, Merck, KGaA, Darmstadt, Germany). Chemiluminescence signals were recorded on a FUJIFILM LAS-3000 and data quantified with the software MultiGauge V3.0. Full size scans of western blots can be found in Supplementary Fig. 8.

**Electron microscopy.** For platinum replica EM, samples were extracted for 3 min in the buffer containing 1% Triton X-100, 2% PEG (MW 35,000), $2\,\mu M$ unlabelled phalloidin (# P2141, Sigma), and $2\,\mu M$ paclitaxel (#T7402, Sigma) in PEM buffer (100 mM PIPES, pH 6.9; 1 mM EGTA; 1 mM MgCl2). For removal of actin filaments, detergent-extracted unfixed cells were incubated with $0.4\, mg\, ml^{-1}$ gelsolin (gift of Dr A. Weber, University of Pennsylvania) in buffer G (pH 6.3, 50 mM MES-KOH, 0.1 mM CaCl2, 2 mM MgCl2, 0.5 mM DTT) for 15 min at room temperature, rinsed in PEM and fixed with 2% glutaraldehyde in the 0.1 M sodium cacodylate buffer, pH 7.3 For immunogold staining of NMII (rabbit polyclonal #BT-564, Biomedical Technologies Inc), detergent-extracted and gelsolin-treated cells were incubated with the primary antibody diluted 1:50 in PEM buffer supplemented with $2\,\mu M$ paclitaxel for 15 min, rinsed in PEM and fixed with 0.2% glutaraldehyde in the sodium cacodylate buffer. Samples were washed with PBS, quenched with $2\, mg\, ml^{-1}$ NaBH4 for 10 min and stained with secondary antibody conjugated with 18 nm colloidal gold (goat anti-rabbit IgG, (#111-215-144, Jackson Immunoresearch Laboratories, Inc, West Grove, PA, USA) at 1:5 dilution in the buffer A (20 mM Tris-HCl (pH8), 0.5M NaCl, 0.05% Tween 20 and 1% BSA). After washing with the same buffer but containing 0.1% BSA, samples were fixed sequentially with 2% glutaraldehyde in sodium cacodylate buffer, 0.1% tannic acid in water, 0.2% uranyl acetate in water, dehydrated through graded ethanols, critical point dried and coated with platinum and carbon. For further details, see ref. 57. Samples were analysed using JEM 1011 transmission EM (JEOL, Peabody, MA, USA) operated at 100 kV. Identification of gold particles in replica EM samples was performed at high magnification after contrast enhancement to distinguish them from other bright objects in the samples.

**Atomic force microscopy.** For AFM, HUVECs were transfected with pEF-C3 and LifeAct-GFP (Ibidi) using BioRad electroporation, and were then grown on 35 mm

glass bottom WillCo-dish plates (WillCo Wells B.V.) coated with gelatin. Experiments were performed the following day at 37 °C on a commercial stand-alone AFM (NanoWizard3, JPK Instruments) combined with an inverted optical microscope (AxioObserver.Z1, Zeiss) driven by JPK NanoWizard software 6.0 and Zen Blue 2012. AFM was operated in Quantitative Imaging mode in proper HUVEC imaging buffer using Olympus BioLever mini with a nominal spring constant of 0.09 N m$^{-1}$. Spring constant was determined using the thermal method of the JPK software before each experiment. We monitored by bright field and fluorescence cells showing TEMs and engaged the AFM tip and scanned as TEM stabilized. Using the QI mode, we were able to perform $64 \times 64$ pixel force mapping scans in less than a minute (or $128 \times 128$ in less than 5 min), compatible with the macroaperture time limit. We applied a force trigger of 0.5–2 nN with a 1–2 μm z ramp, at a 100–250 μm s$^{-1}$ speed. Elasticity and elasticity tomograms were processed using in-house developed software. Elasticity was calculated as Young's modulus using the complete indentation curve at a max indentation of 120–180 nm depending on sample indentation, using the Hertz model corrected for the height of the cell[58], as follows

$$F = \frac{16E}{9} R^{\frac{1}{2}} \delta^{\frac{3}{2}} [1 + 1.133X + 1.283X^2 + 0.769X^3 + 0.0975X^4]$$

F: Force, E: Young's modulus, R: Tip radius, δ: indentation, X : $\delta/h$ with h: sample thickness. We defined the ring region of interest (ROI) as the pixels showing the ring, and the cell ROI as the [1-2] μm band region beyond the ring. Bar plot and significance test (t-test paired of medians, $n = 7$, $P = 0.0009$) was performed using Graphpad Prism 6. Elasticity tomograms were calculated using 20 nm indentation steps starting from the point of contact and calculated using the Hertz formula[59], as follows:

$$F = \frac{4E}{3(1-v^2)} R^{\frac{1}{2}} \delta^{\frac{3}{2}}$$

v: Poisson's ratio. Fluorescence images were acquired with z-stack and reconstructed using focus stacking in Fiji software[60].

**Photoablation and FRAP experiments.** Laser ablation experiments were performed in HUVECs expressing LifeAct-GFP. These experiments were performed with an Inverted Laser Scanning Confocal LSM710NLO (Carl Zeiss). Laser ablations were performed with a two-photon laser-type scaled to 805 nm with pulse width < 100 fs (30 iterations × acquisition every 2 s). Images were processed with Image-J and QuickTime pro 7 (Apple). For FRAP experiments, HUVECs were treated with control- or ezrin-siRNA. After 24 h, EGFP-actin was transfected and incubated for further 24 h, followed by intoxication with C3 for more than 12 h. Then, FRAP experiments were conducted using Leica TCS SP5 II confocal microscopy with HCX PL Apo 63 × /1.2 W CORR objective and Ar 488 nm laser at 37 °C with 5% $CO_2$. Actin cables at the TEM edge were bleached with a circular ROI, and the recovery of the fluorescence intensity of EGFP-actin was monitored. Control and ezrin knockdown cells (10 cells in each group) were analysed using ImageJ software, and the average fluorescence intensity ± s.e.m. of EGFP-actin was plotted. The prebleach value was normalized to 1.

**Statistical analysis.** Data are presented as means ± s.e.m. unless otherwise indicated. The normality of each data set was first assessed using Kolmogoro–Smirnov test. For normally distributed data, unpaired, two-sided Student's t-test was used. For non-normally distributed data, unpaired, two-sided Mann–Whitney test was used. When assessing multiple groups, one-way ANOVA was utilized with Bonferroni post-hoc. P-value for *$P < 0.05$, **$P < 0.01$, ***$P < 0.001$ and ****$P < 0.0001$ were considered statistically significant. The statistical software used was Prism 5.0b (GraphPad Software, San Diego, CA, USA)

**Data availability.** The data that support the findings of this study are available from the corresponding authors upon reasonable request.

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

## Acknowledgements

We thank A. Bongiovanni and T. Bornschlögl for advices and technical support and F. Brochard-Wyart for insightful discussions. We acknowledge the imaging core facilities at the Institut Curie ([PICT-IBiSA] member of the France-BioImaging national research infrastructure (ANR-10-INSB-04)) and the C3M (MICA). Work in E.L. laboratory is financed by INSERM, the 'Investments for the Future' LABEX SIGNALIFE ANR-11-LABX-0028-01. E.L., F.L. and P.B. acknowledge funding from Agence Nationale de la Recherche (ANR-15-CE18-0016-01). The P.B. group belongs to the CNRS consortium CellTiss, to the Labex CelTisPhyBio (ANR-11-LABX0038), and to Paris Sciences et Lettres (ANR-10-IDEX-0001_02). T.S. acknowledges funding from NIH (R01 grant GM-095977). F.L. acknowledges the 'Investments for the Future' EquipEX ImagInEx BioMed ANR-10-EQPX-0004-1 and the BioImaging Center Lille for access to instruments.

## Author contributions

Conception and design was done by E.L., P.B., D.G.-R., F.L., P.L., C.S., T.S., O.C.-E. Acquisition of data was done by C.S., Y.S., A.D., N.E., J.L., D.H., M.P.M., C.P., S.J. and M.C.T. Analysis and interpretation of data was done by all authors. Drafting the article was done by E.L., P.B., D.G.-R. and C.S. All authors have revised and accepted the manuscript.

## Additional information

**Competing interests:** The authors declare no competing financial interests.

