## [Peer Review File · Nature Communications]

Reviewers' comments:

Reviewer #1 (Remarks to the Author):

This manuscript reports an interesting combination of experimental and theoretical work, which shows new evidence for the presence of an actomyosin cable that builds up line tension to stabilize transendothelial cell macroapertures. The paper starts by presenting convincing experimental evidence for the presence of this cable, then puts these findings in the context of a dewetting model that proposes testable predictions, and then closes the circle by testing these predictions experimentally. Altogether, the evidence that ezrin and nonmuscle myosin play a crucial role in stabilizing the actomyosin cable is rather convincing. I nevertheless have a few comments that should be addressed.

Main points:

- I find the title very misleading: ezrin does not "power" line tension; the term "powering" suggests an active, ATP-dependent process. Also myosin does not "power" line tension: the authors actually claim that NMIIA acts by crosslinking, not by contractility. This confusion also comes back in the main text and figure captions.
- Figure 4E (right) suggests after ablation in controls the 'steady state' is transient and R returns to a final value identical to the initial value. Is this an exception, or do all the other curves that are shown only display a limited time window? Is it possible that the maximum area is not an equilibrium value (as assumed throughout the entire article and essential for the proposed mechanism), but a competition between opening and closing?
- The changes induced by depletion of NMIIA and ezrin are attributed to changes in the ring stiffness. However, alternatively these proteins could also affect membrane tension. Measurements of actomyosin ring stiffness (as in figure 1B) upon manipulation of ezrin and NMIIA should be performed to disentangle these contributions.
- The effect of ezrin should be discussed in more detail: is it purely caused by recruitment of actin, making the ring stiffer? Or is there more to it, such as an increase in membrane-actin friction?
 - On page 10 it is argued that it is reasonable to assume 'Constant tension' (lines 11-19). However, the important parameter deciding whether tension is constant is ΔA during ablation (so TEM area before versus after) compared to total area of the cell. Using 50, 12 and 10 micrometers for radii of the cell, TEM after ablation and TEM before ablation respectively, one finds a relative membrane area reduction of 0.4%. The assumption that this amount is negligible is non-obvious as membranes are only extensible by a few percent.
- On Page 9, line 10, it is claimed that the replenishment time is 10 s for F-actin; the data to support this are missing. It would furthermore be instructive to compare this time scale to data on actin turnover time without laser ablation (e.g. by FRAP on actin).
- The last sentence of the abstract claims that ezrin controls tension homeostasis. This claim is unjustified as tension is not measured with and without ezrin. This statement is especially strange as it is claimed that TEM size has a negligible effect on tension. Finally, if ezrin indeed controls tension homeostasis, then it is actually unclear whether ezrin plays this role in the ring or in the surrounding cortex.

Minor comments:

- In general, the paper is well written, however from page 6-7 onwards the paper becomes hard to read due to (admirably) meticulous consideration of alternatives and a few language issues. For instance on Page 7: Alternative hypotheses for stabilization after ablation (remaining cable bending, cable reorganization and importance of convection) mentioned in the text could be removed to SI, since it currently takes the flow out of the story and the alternatives felt unlikely anyway. More suggestions are given below.
- Page 6 line 21: it is unclear how taking into account the energy cost of bending of the membrane puts a limited size to a TEM -- it can put a critical size to it, above which it grows indefinitely and below which it shrinks, but not a steady state size. This should be better explained, or removed.
- Page 14 line 2: remove "slightly".
- Figure 1a: the last stage of liquid dewetting shows many small droplets. This should be changed

to one big droplet as the point is liquid dewetting should not show limited size.

- Figure 1B: replace the rainbow as heatmap by a gradient continuous (1-color) gradient.
- Page 8 line 12: unclear what "membrane wave extensions" mean; please refer to earlier work or add an image in SI that shows what you mean.
- Page 5 line 14-16: rewrite the sentence to make it easier to read. "Our previous findings that (...) stress the importance".
- Page 8 line 19, what do you mean by an "analytic system"?

Reviewer #2 (Remarks to the Author):

In the present study, Stefani et al. investigate biophysical aspects of transendothelial cell macroaperture (TEM) tunnel formation. TEMs control barrier functions of endothelium, the epithelium that lines the interior surface of blood vessels and lymphatic vessels. TEMs represent transient structures whose abnormal size or persistence can provoke vascular leakage. Deciphering the mechanisms that control the size of TEMs in cultured human endothelial cells, the authors revealed the existence of two distinct F-actin networks along the edges of TEMs: branched and bundled actin populations. Branched actin assembly has been previously implicated in the closure of TEMs. While the existence and the role of bundled actin in TEM formation and dynamics remained elusive. Utilizing a combination of optical and electron microscopy imaging with biophysical tools and theoretical modeling, Stefani et al. show that during the opening phase of TEM formation a stiff actomyosin bundle encircles TEMs, limiting their expansion. The exact mechanism for the assembly of the circumferential system of actomyosin bundles during TEM formation is not clear, but the authors illuminate ezrin, the linker protein connecting F-actin to the plasma membrane, and non-muscle myosin IIA, the actin-based molecular motor bundling actin filaments and producing contraction in the actin cytoskeleton, as key molecular players. Based on imaging and functional interrogation approaches, Stefani et al. demonstrate that ezrin acts upstream of myosin IIA by stabilizing cortical actin filaments along the edges of TEMs and thus providing substrates for myosin IIA binding and actomyosin bundle formation. These are novel and interesting observations obtained in a physiologically relevant experimental system using appropriate experimental approaches and theoretical considerations. The results provide new essential pieces of information that can be used in future efforts to build a complete mechanistic model of TEM formation and dynamics. Therefore, overall, the work by Stefani et al. is appropriate for the publication in Nature Communications after additional improvements that are outlined below:

1) Although the role of membrane tension is considered by the authors in the theoretical model, no experimental assessment is made to see how modulation of membrane tension during the process of TEM formation can affect TEM dynamics along with kinetics and hierarchy of recruitment to TEM edges of the identified molecular components. Experiments with deoxycholate, concanavalin A, or other modulators of membrane tension are technically feasible and can reveal or exclude membrane tension as a mechanoresponsive mechanism for the assembly of the specific subset of actin filaments limiting TEM formation.

2) The authors describe two types of F-actin structures co-existing along the edges of TEMs: branched and bundled actin populations. These two actin pools are known to mutually affect each other in a myosin-dependent fashion: high levels of myosin-based contractility convert branched actin into bundled structures, while moderate levels of myosin-based contraction enable the formation of branched and bundled actin structures that could self-segregate into two spatially and functionally distinct domains; low levels of contractility permit mostly branched actin assembly (Lomakin et al. Nature Cell Biology, 2015 [PMID: 26414403]). Highly sensitive to myosin activity, these transitions in F-actin microscopic structure and large-scale organization could contribute to TEM formation/dynamics and thus should be discussed by the authors in the text of the paper.

3) Formins nucleate actin cables prone to bundling. At the same time, formin activity is known to be mechanosensitive (Higashida et al. *Nature Cell Biology*, 2013 [PMID: 23455479]). Therefore, it is important to determine whether formins get recruited to TEM edges (e.g. in response to membrane tension) before actin/ezrin, and whether formin inhibition affects the assembly of the actomyosin bundle encircling TEMs. GFP-mDia1/2 imaging in live cells during TEM formation and treatment of cells with SMIFH2 to inhibit formin-dependent actin nucleation would be sufficient.

We appreciate the favorable comments of the referees as well as their very constructive and helpful suggestions. We addressed carefully all points of concern in the revised manuscript and within this point-by-point response below. All changes made in the revised version of our manuscript are marked in blue.

Reviewer #1 (Remarks to the Author):

This manuscript reports an interesting combination of experimental and theoretical work, which shows new evidence for the presence of an actomyosin cable that builds up line tension to stabilize transendothelial cell macroapertures. The paper starts by presenting convincing experimental evidence for the presence of this cable, then puts these findings in the context of a dewetting model that proposes testable predictions, and then closes the circle by testing these predictions experimentally. Altogether, the evidence that ezrin and nonmuscle myosin play a crucial role in stabilizing the actomyosin cable is rather convincing. I nevertheless have a few comments that should be addressed.

Main points:

- I find the title very misleading: ezrin does not “power” line tension; the term “powering” suggests an active, ATP-dependent process. Also myosin does not “power” line tension: the authors actually claim that NMIIA acts by crosslinking, not by contractility. This confusion also comes back in the main text and figure captions.

We agree with the referee's concern. Accordingly, we have changed the title into « Ezrin enhances line tension along transcellular tunnel edges via NMIIa driven actomyosin cable formation ». We also have modified the legend titles of Figs. 4 and 5: "NMIIa controls TEM size ", " Ezrin controls TEM size ". Similarly, we have changed paragraph titles in the text : P. 5 "we provide evidence of a hierarchical function of ezrin and NMIIa at the edges of TEMs to structure an F-actin bundle and to enhance line tension that ..." , P.16 " NMII limits TEM enlargement by enhancing line tension."

- Figure 4E (right) suggests after ablation in controls the ‘steady state’ is transient and R returns to a final value identical to the initial value. Is this an exception, or do all the other curves that are shown only display a limited time window? Is it possible that the maximum area is not an equilibrium value (as assumed throughout the entire article and essential for the proposed mechanism), but a competition between opening and closing?

In control conditions there are two distinct time scales of TEM dynamics. TEM opening, driven by membrane tension, occurs at short time scales (a minute or less after ablation). The appearance of membrane waves, which cover TEMs, occurs only after the TEM maximum size and the plateau value have been reached, typically after 2 minutes or more; then, closure takes place over several minutes. Thus, the two phenomena, opening and closure, are separate, and we can thus consider that the maximum TEM size reached between opening and closure corresponds to an equilibrium value. We have included this discussion in the revised manuscript (page 12). We have also modified the description of

results to clarify this point see below (or text page 16) :

Clarification page 12 “Since TEM opening occurs at shorter time scales distinct from the beginning of TEM closure driven by lamellipodia-like membrane extensions, we can consider the two phenomena as temporally separated, and thus consider that the maximum TEM size corresponds to an equilibrium value. »

Description of results page 16 “Quantitative analysis of the variation of the radii over time after ablation revealed a critical implication of NMII in limiting their enlargement. Indeed, in NMII-knocked down cells, TEMs continued to grow after ablation at a constant speed over the study period of 350 seconds, while TEMs in control cells underwent closure (Fig. 4e). Pursuing the analysis further revealed that 50% of TEMs were already closed at 500 seconds in control cells (n=6), while 80% of TEMs in siNMII treated cells continued to grow (n=6).»

- The changes induced by depletion of NMIIA and ezrin are attributed to changes in the ring stiffness. However, alternatively these proteins could also affect membrane tension. Measurements of actomyosin ring stiffness (as in figure 1B) upon manipulation of ezrin and NMIIA should be performed to disentangle these contributions.

The referee is right in that variations in ring stiffness might be reflected into variations of Young's modulus, as measured by AFM. We note however that stiffness and tension are two independent, and in general uncorrelated, physical properties. As discussed in the paper, the existence of line tension is evidenced by the opening of the tunnel after laser ablation. Following the referee's remark, we have performed new AFM measurements in different conditions of RNAi treatment followed by cell intoxication. We found no significant differences in Young's modulus between the different conditions (Figure A below). This is consistent with our model, which ascribes an effect of siEzrin/siNMII treatment to changes in line tension, but not in Young's modulus. The question remains as to whether the dominant effect of the si-treatments is to modify line tension or to modify membrane tension (at least one of the two must be modified, to explain the observed effect in tunnel size). To address this point, we have quantified the cell spreading area in different treatment conditions, since spreading depends on surface tension but not on line tension. No significant variations in cell areas are observed (Figure B below), which supports our conclusion that membrane tension is not significantly changed by the treatments. All the data presented in our manuscript taken collectively point toward a local effect of Ezrin and NMII for the control of line tension.

Figure A: Ring elasticity in siCtrl, siEzrin and siNMIIa-depleted cells and intoxicated with ExoC3

Each dot of the scatter plot represents the median of Young's moduli of a ring. Bars represent means +/- SEM, ns $p \geq 0.01$, Welch t-test. Experiments were performed on a JPK NanoWizardIII AFM mounted on a Zeiss AxioVert optical microscope. Cells were monitored by bright-field and when a macroaperture formed and stabilized AFM was engaged to perform force measurements. We calibrated Olympus Biolever mini before each experiment using the SADER method using JPK software 6.

Measurement were performed in QI mode with the following parameters : tip velocity 150

$\mu\text{m/s}$, ramp size 1200 nm, force trigger 500 pN. Scans covered a surface of $5 \mu\text{m}^2$ on 80×80 pixels. Data were analyzed using in-house software for the automatic fitting of Hertz model with the bottom effect correction. The ring elasticity was measured by drawing a region of interest above the ring on each scan.

Figure B: Cell area of siCtrl, siEzrin and siNMIIa-depleted cells and intoxicated with ExoC3

Cells treated with the different siRNA and intoxicated 15 hours with ExoC3. Cells were then stained with phalloidin. Images were taken with Evos fl and area of each TEM measured with Fiji ImageJ. Each value is represented together with histogram of means \pm SEM ($n=90$ cells, each condition with 2 independent replicas), $p \geq 0.01$ Mann-Whitney test.

- The effect of ezrin should be discussed in more detail: is it purely caused by recruitment of actin, making the ring stiffer? Or is there more to it, such as an increase in membrane-actin friction?

Our measurements, together with our biophysical model, show that the presence of ezrin increases the tension of the cable, which contributes to reduce TEM size, thus implying that ezrin increases the stiffness (i.e., the ratio between force and deformation) of the cable. This conclusion is further supported by our FRAP analysis showing that ezrin stabilizes actin around TEMs. We cannot exclude an additional effect of ezrin on membrane-actin friction. We note however that, if ezrin had a significant effect on membrane-actin friction, we would expect to observe an effect of ezrin silencing on TEM opening dynamics, which is not the case (see below figure at short time scale from data in Fig. 5e). Therefore, we can conclude that ezrin essentially contributes to form a stiff cable, whereas its effect on membrane-actin friction, if it exists, is not discernible from our data. As requested by the referee, a more detailed discussion of this point has been included in the revised manuscript (page 21).

Figure C : TEM opening dynamics in siCtrl and siEzrin depleted-cells and intoxicated with ExoC3

Graph depicts at short time scale after ablation the increase of TEM radius in HUVECs treated with siRNA control (siCtrl) or siRNA ezrin (siEzrin) and intoxicated 24 h with exoC3. Data are represented as means, $n > 5$.

- On page 10 it is argued that it is reasonable to assume 'Constant tension' (lines 11-19). However, the important parameter deciding whether tension is constant is ΔA during ablation (so TEM area before versus after) compared to total area of the cell. Using 50, 12 and 10 micrometers for radii of the cell, TEM after ablation and TEM before ablation respectively, one finds a relative membrane area reduction of 0.4%. The assumption that this amount is negligible is non-obvious as membranes are only extensible by a few percent.

The Reviewer is right in that the intrinsic membrane extensibility is low, due to a high stretching modulus, as measured for example by Hochmuth, Mohandas and Blackshear, *Biophys J.*, 1973. However, it is also observed that membrane surface can vary significantly, by about 10% or beyond, for example during cell spreading. This apparent contradiction can be explained by the existence of membrane reservoirs in the form of caveolae, invaginations/evaginations, as well as by active exocytic processes (as discussed for example by Kosmalska et al., *Nature Comm.* 6, 7292, 2015). Changes in surface tension due to changes in apparent membrane surface area can be quantified using Helfrich's law, as mentioned in our article. For a change of membrane surface area of 0.4%, Helfrich's law predicts a change in surface tension of 1%, which is negligible, thus supporting our model's assumption that surface tension is approximately constant. Moreover, we observe that spontaneous opening of a TEM can be followed by opening of other TEMs in the same cell before the first TEM has closed back. The opening dynamics of the first and of subsequent TEMs are similar, which suggests that the driving force (membrane tension) must remain similar. This observation supports our conclusion that the value of membrane tension is essentially unaffected by TEM opening. These aspects are discussed on page 11.

- On Page 9, line 10, it is claimed that the replenishment time is 10 s for F-actin; the data to support this are missing. It would furthermore be instructive to compare this time scale to data on actin turnover time without laser ablation (e.g. by FRAP on actin).

We thank the referee for this remark. Data were presented in Figure 6d only and therefore we now discuss directly the results in the text also referring to figure 2c and 2e, see below or on page 9 and discussion page 20.

Page 9, line 11 :

« Analysis of LifeAct-GFP signal along enlarged TEMs allowed us to calculate a F-actin replenishment time in the expending zone (as shown in Fig. 2c, arrow) of 10.1 seconds \pm 1.6 SEM, $n=14$ TEMs, concomitantly with the TEMs reaching the new equilibrium radius, R_{eq} (Fig. 2e). »

Page 20, line 20 :

"The FRAP analysis revealed that the depletion of ezrin leads to a 1.3-fold increase in the recovery rate of F-actin (Ctrl $t_{1/2} = 4.6$ s versus siEzrin $t_{1/2} = 3.6$ s) and to a 1.4-fold increase in the size of mobile fraction (Fig. 6e). "

- The last sentence of the abstract claims that ezrin controls tension homeostasis. This claim

is unjustified as tension is not measured with and without ezrin. This statement is especially strange as it is claimed that TEM size has a negligible effect on tension. Finally, if ezrin indeed controls tension homeostasis, then it is actually unclear whether ezrin plays this role in the ring or in the surrounding cortex.

We thank the referee for her/his pertinent remark. We have changed the sentence by: « Collectively, our findings ascribe to ezrin and NMIIa a critical function of enhancing line tension at the cell boundary surrounding the TEMs by promoting the formation of an actomyosin ring.»

Minor comments:

- In general, the paper is well written, however from page 6-7 onwards the paper becomes hard to read due to (admirably) meticulous consideration of alternatives and a few language issues. For instance on Page 7: Alternative hypotheses for stabilization after ablation (remaining cable bending, cable reorganization and importance of convection) mentioned in the text could be removed to SI, since it currently takes the flow out of the story and the alternatives felt unlikely anyway. More suggestions are given below.

Following the Reviewer's suggestion, the aforementioned discussion has been moved to the Supplementary Information (see Supplementary Note 1).

- Page 6 line 21: it is unclear how taking into account the energy cost of bending of the membrane puts a limited size to a TEM -- it can put a critical size to it, above which it grows indefinitely and below which it shrinks, but not a steady state size. This should be better explained, or removed.

The energy cost of bending the membrane yields to a constant value of line tension, which, combined to a reduction of membrane tension as the TEM opens, can indeed explain the existence of a steady-state TEM size. This hypothesis was assumed in our previous modeling article on TEM opening (see Gonzalez-Rodriguez et al., Phys Rev Lett 2012), and the mathematical modeling showing how this hypothesis leads to prediction of an equilibrium TEM size can be found there. The Reviewer is however right in that the membrane-bending line tension also puts a critical size to TEMs, as discussed in our 2012 article as well.

We have revised our manuscript to explicitly refer the reader to this previous article for details on the hypothesis of a bending-energy origin of the line tension (see modification on pages 6 line 21).

“This hypothesis, together with a membrane tension that decreases with TEM size, was used in our earlier model of TEM dynamics to explain the equilibrium size of TEMs formed de novo⁵. »

As also explained in the manuscript, this simple energetic hypothesis does not suffice to explain the TEM ablation experiments, this is why the cable-induced component of the line tension must be taken into account to explain the new experimental data.

- Page 14 line 2: remove “slightly”.

Done

- Figure 1a: the last stage of liquid dewetting shows many small droplets. This should be changed to one big droplet as the point is liquid dewetting should not show limited size.

Showing one big droplet as the final picture of liquid dewetting would not correspond to the physical reality of viscous liquid dewetting. Indeed, liquid dewetting does not yield one single, connected body of liquid, but rather many small droplets as originally shown. However, the referee is right in that this last stage of liquid dewetting is no longer analogous to cellular dewetting, where no "cellular droplets" are formed. Therefore, Figure 1a has been modified (see below) to show only the stages of liquid dewetting that are analogous to cellular dewetting, and to graphically illustrate the fact that liquid dewetting does not reach a maximal size.

- Figure 1B: replace the rainbow as heatmap by a gradient continuous (1-color) gradient. Heat map has been replaced.

- Page 8 line 12: unclear what "membrane wave extensions" mean; please refer to earlier work or add an image in SI that shows what you mean.

The term membrane wave extensions is now introduced into the discussion of data corresponding to Figure 1 and we refer to movie 1 (page 7, line 24).

"...a previously characterized branched network involved in lamellipodia-like membrane wave extensions that invade the hole for closure (Movie 1)".

On page 8, line 12, we also changed the sentence into:

« These experiments were performed on stable TEMs that, at the moment of the ablation, did not display membrane wave extensions engaged in the closure of the hole (movie 1). »

- Page 5 line 14-16: rewrite the sentence to make it easier to read. "Our previous findings that (...) stress the importance".

This sentence has been changed into:

Page 5, line 15: « Our previous findings that unrestricted widening of TEMs leads to a lethal increase of vascular permeability stress the importance of a cell autonomous control of this phenomenon. »

- Page 8 line 19, what do you mean by an "analytic system"?

This sentence has been changed to:

Page 8, line 20: « The laser ablation technique allows quantification of TEM enlargement as a function of their initial size »

Reviewer #2 (Remarks to the Author):

In the present study, Stefani et al. investigate biophysical aspects of transendothelial cell

macroaperture (TEM) tunnel formation. TEMs control barrier functions of endothelium, the epithelium that lines the interior surface of blood vessels and lymphatic vessels. TEMs represent transient structures whose abnormal size or persistence can provoke vascular leakage. Deciphering the mechanisms that control the size of TEMs in cultured human endothelial cells, the authors revealed the existence of two distinct F-actin networks along the edges of TEMs: branched and bundled actin populations. Branched actin assembly has been previously implicated in the closure of TEMs. While the existence and the role of bundled actin in TEM formation and dynamics remained elusive. Utilizing a combination of optical and electron microscopy imaging with biophysical tools and theoretical modeling, Stefani et al. show that during the opening phase of TEM formation a stiff actomyosin bundle encircles TEMs, limiting their expansion. The exact mechanism for the assembly of the circumferential system of actomyosin bundles during TEM formation is not clear, but the authors illuminate ezrin, the linker protein connecting F-actin to the plasma membrane, and non-muscle myosin IIA, the actin-based molecular motor bundling actin filaments and producing contraction in the actin cytoskeleton, as key molecular players. Based on imaging and functional interrogation approaches, Stefani et al. demonstrate that ezrin acts upstream of myosin IIA by stabilizing cortical actin filaments along the edges of TEMs and thus providing substrates for myosin IIA binding and actomyosin bundle formation. These are novel and interesting observations obtained in a physiologically relevant experimental system using appropriate experimental approaches and theoretical considerations. The results provide new essential pieces of information that can be used in future efforts to build a complete mechanistic model of TEM formation and dynamics. Therefore, overall, the work by Stefani et al. is appropriate for the publication in Nature Communications after additional improvements that are outlined below:

1) Although the role of membrane tension is considered by the authors in the theoretical model, no experimental assessment is made to see how modulation of membrane tension during the process of TEM formation can affect TEM dynamics along with kinetics and hierarchy of recruitment to TEM edges of the identified molecular components. Experiments with deoxycholate, concanavalin A, or other modulators of membrane tension are technically feasible and can reveal or exclude membrane tension as a mechanoresponsive mechanism for the assembly of the specific subset of actin filaments limiting TEM formation.

Following the Reviewer's suggestion, we have performed experiments where cells were treated with deoxycholic acid. Upon cell treatment with a 300mM deoxycholic acid for one hour, the maximum TEM size exhibits a significant decrease: average maximum TEM area of $47 \mu\text{m}^2$ versus $178 \mu\text{m}^2$ in control conditions. This result is consistent with membrane tension being the driving force in TEM opening. Indeed, our theoretical model predicts a decrease in maximum TEM size with decreasing membrane tension. We note however that the precise manner in which deoxycholate perturbs the membrane is unclear. Rather than affecting membrane tension, deoxycholic acid may induce a spontaneous curvature of the membrane upon insertion, which would lead to a decrease of the line tension with formation of larger TEMs (see for instance, our paper Gonzalez-Rodriguez et al., Phys Rev Lett 2012 and our answer to referee 1 above). As already discussed in the current manuscript, the contribution of membrane curvature to line tension and TEM maximum size is not dominant compared to cable-induced tension. Thus, these new experiments with deoxycholic acid tend to confirm the major contribution of line tension, and certainly of

membrane mechanics. Due to space limitation and to the main focus of our paper, we prefer not to discuss the different interpretations of these experiments in the manuscript and rather conclude that the new deoxycholic acid experiments support our hypothesis that membrane mechanics plays a crucial role in TEM opening. We have thus added these new observations on page 10 in the revised manuscript:

Page 10, line 1: « In support of a crucial role of plasma membrane mechanics in TEM opening, membrane perturbation by treating cells with the detergent deoxycholic acid (Raucher and Sheetz 2000 J Cell Biol) reduced the mean area of TEMs by 3.77 fold (Supplementary Fig. 1). »

2) The authors describe two types of F-actin structures co-existing along the edges of TEMs: branched and bundled actin populations. These two actin pools are known to mutually affect each other in a myosin-dependent fashion: high levels of myosin-based contractility convert branched actin into bundled structures, while moderate levels of myosin-based contraction enable the formation of branched and bundled actin structures that could self-segregate into two spatially and functionally distinct domains; low levels of contractility permit mostly branched actin assembly (Lomakin et al. Nature Cell Biology, 2015 [PMID: 26414403]). Highly sensitive to myosin activity, these transitions in F-actin microscopic structure and large-scale organization could contribute to TEM formation/dynamics and thus should be discussed by the authors in the text of the paper.

We thank the referee for this very insightful comment. We have added a paragraph along these lines in the revised manuscript in the discussion, p. 21-22 :

Page 21, line 22 :

“The extent of myosin-based contractility controls the switch between branched and bundled actin-based structures that plays a critical role in tissue organization versus cell polarization and motility⁴⁴. Formation of actomyosin cables along TEMs occurs in a peculiar cellular background of low RhoA-driven myosin contractility and membrane retraction. Thus, the low myosin contractility favors the formation of dendritic actin network, which is likely compensated by a high compression along TEMs due membrane receding that is responsible for a local increase of F-actin density and bundling, thereby allowing both actin structures to coexist in condition of low RhoA activity.»

3) Formins nucleate actin cables prone to bundling. At the same time, formin activity is known to be mechanosensitive (Higashida et al. Nature Cell Biology, 2013 [PMID: 23455479]). Therefore, it is important to determine whether formins get recruited to TEM edges (e.g. in response to membrane tension) before actin/ezrin, and whether formin inhibition affects the assembly of the actomyosin bundle encircling TEMs. GFP-mDia1/2 imaging in live cells during TEM formation and treatment of cells with SMIFH2 to inhibit formin-dependent actin nucleation would be sufficient.

We thank the referee for this important point. Indeed, Higashida et al., 2013 Nat Cell Biol have elegantly shown that mDIA1 is a crucial mechanosensitive actin nucleator. However, they also showed that C3 treatment abolishes the mechanosensitive actin nucleation property of full-length mDIA1, whereas the FH2 fragment still responds to mechanical action.

We anticipated that mDIA1 activity might be negligible when RhoA is markedly reduced in intoxicated cells. Nevertheless, following referee's suggestion, we have performed experiments to analyze mDIA1 localization at TEM edges. We did not detect a significant localization of mDIA1 around the TEMs in EDIN-treated cells (Figure D).

We also think that it is an important point to discuss in the revised version of the manuscript See page 23, line 7:

“Although the mDIA1 formin displays a mechanosensitive actin nucleation activity when cell tension is released⁴⁹, we did not detect its accumulation at TEM edges.»

Figure D : mDia1 localization in exoC3 treated cells

HUVECs were transfected with EGFP-mDia1 expressing vector. Transfected cells are treated with exoC3 toxin at 50µg/ml for 18 hours. Cells are fixed and stained with phalloidin-TRITC. Images are taken with Nikon A1R confocal microscope. Scale bar, 10µm.

We have also tested the effect of SMIFH2 mDIA inhibitor and did not observe an increase of TEM area that would result from a specific function of mDIA in controlling the formation of the actomyosin cable at the edge of TEMs. Indeed, as shown in figure E, this inhibitor reduced TEM size, which can be attributed to the decrease of cell perimeter induced in these conditions and probably to reduction of cell membrane tension.

Figure E: effect of SMIFH2 mDIA inhibitor on TEM area

HUVEC cells were seeded on coverslips and treated with 50µg/ml C3 exotoxin overnight. Cells were then treated with 10µM SMIFH2 for 1 or 3 hours. Cells were then stained with phalloidin. Images were taken with Leica DM5500 and area of each TEM measured with Fiji ImageJ. Values correspond to means ± SEM (n>200 TEMs, each condition with 2 independent replicas). Two-tailed student t-test, * p < 0.01.

REVIEWERS' COMMENTS:

Reviewer #1 (Remarks to the Author):

The authors have convincingly responded to all queries of the referees and have significantly improved and clarified their paper. In my opinion, the manuscript in its current form is satisfactory.

Reviewer #2 (Remarks to the Author):

I have carefully examined the revised version of the manuscript and am convinced that in its current form the paper is ready for publication.

- 1) The authors were able to address all the important questions raised by the reviewers.
- 2) The new experiments strengthen the key findings described in the original submission.

Considering the novelty of the work, original experimental & theoretical design, and appropriate statistical analyses, I recommend the paper by Stefani et al. for publication.